# Identification and Estimation of Conditional Average Partial Causal Effects via Instrumental Variable

**Yuta Kawakami**[1,2]  **Manabu Kuroki**[1]  **Jin Tian**[2]

[1]Department of Mathematics, Physics, Electrical Engineering and Computer Science, Yokohama National University, Yokohama, Kanagawa, JAPAN
[2]Department of Computer Science, Iowa State University, Ames, Iowa, USA

## Abstract

There has been considerable recent interest in estimating heterogeneous causal effects. In this paper, we study conditional average partial causal effects (CAPCE) to reveal the heterogeneity of causal effects with continuous treatment. We provide conditions for identifying CAPCE in an instrumental variable setting. Notably, CAPCE is identifiable under a weaker assumption than required by a commonly used measure for estimating heterogeneous causal effects of continuous treatment. We develop three families of CAPCE estimators: sieve, parametric, and reproducing kernel Hilbert space (RKHS)-based, and analyze their statistical properties. We illustrate the proposed CAPCE estimators on synthetic and real-world data.

## 1 INTRODUCTION

Instrumental variable (IV) analysis is a powerful tool used to elucidate causal relationships between treatment ($X$) and outcome ($Y$) when a controlled experiment is not feasible [Imbens, 2014, Angrist and Krueger, 2001]. Traditionally, there are a large number of works focusing on binary or categorical treatment variables [Imbens and Angrist, 1994, Balke and Pearl, 1997]; recently, there has been a growing interest in continuous treatment variables [Hirano and Imbens, 2005, Kennedy et al., 2017, Bahadori et al., 2022]. There is also considerable recent interest in estimating heterogeneous causal effects across subsets of the population [Athey and Imbens, 2016, Ding et al., 2016, Athey and Imbens, 2019, Künzel et al., 2019, Wager and Athey, 2018, Zhang et al., 2022, Singh et al., 2023], including IV-based methods [Angrist, 2004, Syrgkanis et al., 2019, Huntington-Klein, 2020, Bargagli-Stoffi et al., 2022]. Most of the works focus on *conditional average causal effect (CACE)* $\mathbb{E}[Y_1 - Y_0|\boldsymbol{w}]$, also known as conditional average treatment effect (CATE),

for evaluating heterogeneous causal effects of a binary $X$, where $Y_x$ denotes the potential outcome under treatment $X = x$, and $\boldsymbol{W}$ are covariates (e.g. gender, age, and race).

In this work, we study estimating heterogeneous causal effects of a *continuous* treatment via the IV method. Existing work in this direction has focused on estimating $\mathbb{E}[Y_x|\boldsymbol{w}]$. The most widely used methods include parametric two-stage least squared (PTSLS) [Wright, 1928, Angrist and Pischke, 2009, Wooldridge, 2010], sieve nonparametric two-stage least squared (sieve NTSLS) [Newey and Powell, 2003, Chen and Christensen, 2018], and Kernel IV [Singh et al., 2019]. The line of works in [Syrgkanis et al., 2019, Dikkala et al., 2020, Muandet et al., 2020, Bennett et al., 2023] focus on the efficiency of estimators assuming simple additive errors. All these methods rely on a *separability* assumption for identifying $\mathbb{E}[Y_x|\boldsymbol{w}]$ [Newey and Powell, 2003].

Another quantity for evaluating the causal effects of a continuous treatment is average partial causal effect (APCE) $\mathbb{E}[\partial_x Y_x]$ [Chamberlain, 1984, Wooldridge, 2005, Graham and Powell, 2012]. Wong [2022] provided a condition for identifying $\mathbb{E}[\partial_x Y_x]$ and Kawakami et al. [2023] presented APCE estimators.

In this paper, we consider $\mathbb{E}[\partial_x Y_x|\boldsymbol{w}]$, termed *conditional average partial causal effect (CAPCE)*, to capture the heterogeneous causal effects of a continuous treatment. CAPCE extends APCE and is a natural generalization of the CACE of a binary treatment. The quantity represented by CAPCE has been implicitly studied in the literature (e.g. [Galagate, 2016]). Still existing works have focused on $\mathbb{E}[Y_x|\boldsymbol{w}]$. One contribution of this work is to show that under the IV model, CAPCE is identifiable under a *weaker* separability assumption than required by the previous work (sieve NTSLS, PTSLS, Kernel IV) for identifying $\mathbb{E}[Y_x|\boldsymbol{w}]$. Thus, computing CAPCE allows scientists to estimate causal effects in a larger class of models. Granted, given an estimated $\mathbb{E}[Y_x|\boldsymbol{w}]$, one can compute its derivative to obtain CAPCE, but not the other way around. However, in practice, the causal effect from a reference point (e.g., CACE) is often the main inter-

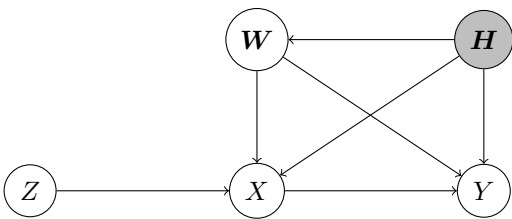

Figure 1: A causal graph representing the IV model.

est, and CAPCE is enough to compute causal effects from a reference point: $\mathbb{E}[Y_{x''} - Y_{x'}|\boldsymbol{w}] = \int_{x'}^{x''} \mathbb{E}[\partial_x Y_x|\boldsymbol{w}]dx$.

We then develop three families of methods for estimating CAPCE: sieve, parametric, and reproducing kernel Hilbert space (RKHS)-based, and analyze their statistical properties. Finally, we illustrate the proposed estimators on synthetic data, showing superior performance to existing methods. We also evaluate CAPCE in a real-world dataset.

## 2 NOTATION AND BACKGROUND

We represent each variable with a capital letter $(X)$ and its realized value with a small letter $(x)$. Let $\Omega_X$ be the domain of $X$, $\mathbb{E}[Y]$ be the expectation of $Y$, $\mathbb{P}(X \leq x)$ be the cumulative distribution function (CDF) of $X$, and $\mathfrak{p}(X = x)$ be the probability density function (PDF) of $X$. A metric space $\langle \Omega, d \rangle$, where distance function $d(x, y)$ is defined by a given norm $\|x - y\|$ for $x, y \in \Omega$, is compact if every sequence in $\Omega$ has a convergent sub-sequence whose limit is in $\Omega$. If every Cauchy sequence of points in $\Omega$ has a limit in $\Omega$, $\Omega$ is called complete.

**Sobolev norm** [Gallant and Nychka, 1987, Leoni, 2009]. Let $\boldsymbol{\lambda}$ be a $d + 1$ dimensional vector of non-negative integer, $|\boldsymbol{\lambda}| = \sum_{l=1}^{d+1} \lambda_l$, and $D^{\boldsymbol{\lambda}} f(x, \boldsymbol{w}) = \partial^{|\boldsymbol{\lambda}|} f(x, \boldsymbol{w}) / \partial x^{\lambda_1} \partial w_1^{\lambda_2} \cdots \partial w_d^{\lambda_{d+1}}$. Sobolev norm is defined as follows: $\|f\|_{W^{l,p}} = \left\{ \sum_{|\boldsymbol{\lambda}| \leq l} \int \{D^{\boldsymbol{\lambda}} f(x, \boldsymbol{w})\}^p dx d\boldsymbol{w} \right\}^{1/p}$ for $1 \leq p < \infty$, and $\|f\|_{W^{l,\infty}} = \max_{|\boldsymbol{\lambda}| \leq l} \sup_{(x,\boldsymbol{w})} D^{\boldsymbol{\lambda}} f(x, \boldsymbol{w})$. Note that $W^{0,p}$ norm coincides with $L_p$ norm for $1 \leq p \leq \infty$.

**Structural Causal Models (SCM).** We use SCM as our framework [Pearl, 2009]. An SCM $\mathcal{M}$ is a tuple $\langle \boldsymbol{V}, \boldsymbol{U}, \mathcal{F}, \mathbb{P}_{\boldsymbol{U}} \rangle$, where $\boldsymbol{U}$ is a set of exogenous (unobserved) variables following a joint distribution $\mathbb{P}_{\boldsymbol{U}}$, and $\boldsymbol{V}$ is a set of endogenous (observable) variables whose values are determined by structural functions $\mathcal{F} = \{f_{V_i}\}_{V_i \in \boldsymbol{V}}$ such that $v_i := f_{V_i}(\mathbf{pa}_{V_i}, \boldsymbol{u}_{V_i})$ where $\mathbf{PA}_{V_i} \subseteq \boldsymbol{V}$ and $U_{V_i} \subseteq \boldsymbol{U}$. Each SCM $\mathcal{M}$ induces an observational distribution $\mathbb{P}_{\boldsymbol{V}}$

over $\boldsymbol{V}$, and a causal graph $G(\mathcal{M})$ over $\boldsymbol{V}$ in which there exists a directed edge from every variable in $\mathbf{PA}_{V_i}$ to $V_i$. An intervention $do(\boldsymbol{x})$ of setting endogenous variables $\boldsymbol{X}$ to constants $\boldsymbol{x}$ replaces the functions of $\boldsymbol{X}$ by the constants $\boldsymbol{x}$ and induces a *sub-model* $\mathcal{M}_{\boldsymbol{x}}$. We denote the potential outcome $Y$ under intervention $do(\boldsymbol{x})$ by $Y_{\boldsymbol{x}}(\boldsymbol{u})$, which is the solution of $Y$ in the sub-model $\mathcal{M}_{\boldsymbol{x}}$ given $\boldsymbol{U} = \boldsymbol{u}$.

**Instrumental Variable (IV) Model with Covariates.** We consider the IV model represented by the causal graph in Fig 1, with the following SCM $\mathcal{M}_{IV}$ over $\boldsymbol{V} = \{Z, X, Y, \boldsymbol{W}\}$ and $\boldsymbol{U} = \{\boldsymbol{H}, \boldsymbol{u}_X, \boldsymbol{u}_Y, \boldsymbol{u}_Z, \boldsymbol{u}_{\boldsymbol{W}}\}$:

$$Y := f_Y(X, \boldsymbol{W}, \boldsymbol{H}, \boldsymbol{u}_Y), \ X := f_X(Z, \boldsymbol{W}, \boldsymbol{H}, \boldsymbol{u}_X),$$
$$\boldsymbol{W} := f_{\boldsymbol{W}}(\boldsymbol{H}, \boldsymbol{u}_{\boldsymbol{W}}), \ Z := f_Z(\boldsymbol{u}_Z), \tag{1}$$

where $f_{\boldsymbol{W}}$ is a vector function. We assume all variables are continuous, $\boldsymbol{W}$ are $d$-dimensional pre-treatment covariates, and $\boldsymbol{H}$ stands for unmeasured confounders. This IV model has been studied in e.g., [Hartford et al., 2017, Huntington-Klein, 2020]. We further consider an IV model with an additional edge $\boldsymbol{W} \to Z$ in Appendix A.2.

**Related work.** Under the IV model, Newey and Powell [2003] introduced sieve NTSLS for identifying and estimating $\mathbb{E}[Y_x|\boldsymbol{w}]$ via an integral equation, $\mathbb{E}[Y|Z = z] = \int_{\Omega_{\boldsymbol{W}}} \int_{\Omega_X} \mathfrak{p}(X = x, \boldsymbol{W} = \boldsymbol{w}|Z = z)\mathbb{E}[Y_x|\boldsymbol{w}]dxd\boldsymbol{w}$, under the following assumption called *separability*:

$$f_Y(X, \boldsymbol{W}, \boldsymbol{H}, \boldsymbol{u}_Y) = f_Y^1(X, \boldsymbol{W}, \boldsymbol{u}_Y) + f_Y^2(\boldsymbol{H}, \boldsymbol{u}_Y),$$
$$\mathbb{E}[f_Y^2(\boldsymbol{H}, \boldsymbol{u}_Y)|\boldsymbol{W}] = 0, \tag{2}$$

which says the function $f_Y(X, \boldsymbol{W}, \boldsymbol{H}, \boldsymbol{u}_Y)$ is in the form of a summation of two functions, one over $(X, \boldsymbol{W})$ and one over $\boldsymbol{H}$. Parametric PTSLS [Angrist and Pischke, 2009, Wooldridge, 2010] and Kernel IV [Singh et al., 2019] methods for estimating $\mathbb{E}[Y_x|\boldsymbol{w}]$ have also been developed under the separability assumption.

Recently, Wong [2022] introduced an integral equation for identifying APCE $\mathbb{E}[\partial_x Y_x] := \mathbb{E}_{\boldsymbol{U}}[\partial_x Y_x(\boldsymbol{U})]$ under the IV model with no covariates $\boldsymbol{W}$: $\mathbb{E}[Y|Z = z] - \mathbb{E}[Y|Z = z_0] = -\int_{\Omega_X} \{\mathbb{P}(X \leq x|Z = z) - \mathbb{P}(X \leq x|Z = z_0)\}\mathbb{E}[\partial_x Y_x]dx$. Kawakami et al. [2023] has developed parametric (P-APCE) and Picard iteration-based (N-APCE) estimators for APCE. In this paper, we extend their results and develop three families of methods for estimating CACPE $\mathbb{E}[\partial_x Y_x|\boldsymbol{w}]$. Our parametric estimator reduces to P-APCE when $\boldsymbol{W}$ is empty. The sieve and RKHS estimators in this paper were not provided in [Kawakami et al., 2023]. We note that Picard-iteration estimator in [Kawakami et al., 2023] is not suitable here because equation (3) uses a PDF in the integral kernel instead of a CDF in the equation for APCE.

# 3 IDENTIFICATION OF CAPCE

First, we formally define *conditional average partial causal effect (CAPCE)* to capture the heterogeneous causal effects of a continuous treatment. Then we present a theorem for identifying CAPCE under the IV model.

**Definition 1** (CAPCE). $\mathbb{E}[\partial_x Y_x | \boldsymbol{w}] := \mathbb{E}_{\boldsymbol{U}}\left[\frac{\partial}{\partial x} Y_x(\boldsymbol{U}) \Big| \boldsymbol{W} = \boldsymbol{w}\right].$

CAPCE is a real-valued function from $x \in \Omega_X$ and $\boldsymbol{w} \in \Omega_{\boldsymbol{W}}$ to $\mathbb{R}$. It is a generalization of CACE for continuous treatment. It is also a generalization of APCE $\mathbb{E}[\partial_x Y_x]$ to represent heterogeneous causal effects. Next, we present conditions for identifying CAPCE under the IV model.

**Assumption 3.1.** *Under the SCM $\mathcal{M}_{IV}$, given $\boldsymbol{W} = \boldsymbol{w}$,*

1. *Instrument relevance: IV $Z$ has a causal effect on $X$, i.e., $\mathbb{E}[X_z]$ is not a constant function of $z$.*

2. *$Y_x$ is differentiable and bounded in $x \in \Omega_X$.*

3. $\sup_{x,z,\boldsymbol{w}} \mathfrak{p}(X_z = x | \boldsymbol{W} = \boldsymbol{w}) < \infty.$

4. *The set of distributions $\mathbb{P}(X | Z = z, \boldsymbol{W} = \boldsymbol{w})$ induced by varying $z$ is a complete set.*

The first assumption is standard for the IV setting. The second assumption means that there exists CAPCE for all subjects for $x \in \Omega_X$ and $\boldsymbol{w} \in \Omega_{\boldsymbol{W}}$. The third assumption means the density function of $X_{z,\boldsymbol{w}}$ is bounded. The fourth assumption implies that $h$ is a zero function if $\mathbb{E}[h(X)|Z = z, \boldsymbol{W} = \boldsymbol{w}]$ does not depend on $z$ for all $\boldsymbol{w} \in \Omega_{\boldsymbol{W}}$, which is also assumed in [Newey and Powell, 2003] for identifying $\mathbb{E}[Y_x | \boldsymbol{w}]$.

**Assumption 3.2** (Separability on $X$). *$f_Y(X, \boldsymbol{W}, \boldsymbol{H}, \boldsymbol{u}_Y)$ is in the form of a summation of two functions over $X$ and $\boldsymbol{H}$ separately, i.e., $f_Y(X, \boldsymbol{W}, \boldsymbol{H}, \boldsymbol{u}_Y) = f_Y^1(X, \boldsymbol{W}, \boldsymbol{u}_Y) + f_Y^2(\boldsymbol{W}, \boldsymbol{H}, \boldsymbol{u}_Y)$.*

We obtain the following result.

**Theorem 3.1** (Identification of CAPCE). *Under SCM $\mathcal{M}_{IV}$ and Assumptions 3.1 and 3.2, CAPCE $\mathbb{E}[\partial_x Y_x | \boldsymbol{w}]$ is identifiable from distributions $\mathbb{P}(X, \boldsymbol{W} | Z)$ and $\mathbb{P}(Y | Z)$ via the integral equation:*

$$\mu(z) = \int_{\Omega_{\boldsymbol{W}}} \int_{\Omega_X} k(z, x, \boldsymbol{w}) \mathbb{E}[\partial_x Y_x | \boldsymbol{w}] dx d\boldsymbol{w}, \quad (3)$$

*where $\mu(z) = \mathbb{E}[Y | Z = z_0] - \mathbb{E}[Y | Z = z], k(z, x, \boldsymbol{w}) = \mathfrak{p}(X \le x, \boldsymbol{W} = \boldsymbol{w} | Z = z) - \mathfrak{p}(X \le x, \boldsymbol{W} = \boldsymbol{w} | Z = z_0)$, and $z_0$ is a arbitrary fixed value.*

**Remark:** Assumption 3.2 is weaker than the assumption (2) needed by existing work sieve NTSLS [Newey and Powell,

2003], PTSLS [Wooldridge, 2010], and Kernel IV [Singh et al., 2019] for identifying $\mathbb{E}[Y_x | \boldsymbol{w}]$, which require both covariates $\boldsymbol{W}$ and the treatment $X$ to be separable from the unmeasured confounders $\boldsymbol{H}$. Assumption 3.2 is particularly less restrictive when there are many covariates. Theorem 3.1 states that *CAPCE $\mathbb{E}[\partial_x Y_x | \boldsymbol{w}]$ is identifiable under a weaker assumption than required by $\mathbb{E}[Y_x | \boldsymbol{w}]$.* The result enables us to compute causal effects in IV models where Assumption 3.2 holds but assumption (2) does not such that existing methods are not applicable. Theorem 3.1 extends the results in [Wong, 2022, Kawakami et al., 2023] for identifying APCE $\mathbb{E}[\partial_x Y_x]$; however, it is worth noting that this important point about weaker separability assumption does not arise in the work of Wong [2022] and Kawakami et al. [2023] because they study the setting with no covariates $\boldsymbol{W}$.

# 4 ESTIMATION OF CAPCE

In this section, we develop three families of methods for estimating CAPCE from data based on Theorem 3.1. We do not need samples from the joint $\mathfrak{p}(Z, X, Y, \boldsymbol{W})$, but rather two datasets $\mathcal{D}^{(1)} = \{x_i^{(1)}, \boldsymbol{w}_i^{(1)}, z_i^{(1)}\}_{i=1}^{N_1}$ and $\mathcal{D}^{(2)} = \{y_i^{(2)}, z_i^{(2)}\}_{i=1}^{N_2}$ known as two-samples IV methods [Singh et al., 2019, Angrist and Krueger, 1992].

## 4.1 SIEVE CAPCE ESTIMATOR

Sieve estimators are a class of non-parametric estimators that use progressively more complex models to estimate an unknown function as more data becomes available [Geman and Hwang, 1982].

**Approximation by Orthonormal Basis Functions.** We approximate the CAPCE $\mathbb{E}[\partial_x Y_x | \boldsymbol{w}]$ by a set of orthonormal basis functions, such as Hermite polynomial functions [Hermite, 2009]. Specifically,

$$\mathbb{E}[\partial_x Y_x | \boldsymbol{w}] \equiv g_0(x, \boldsymbol{w}) \approx g(x, \boldsymbol{w}) = \sum_{j=1}^{J} \beta_j \phi_j(x, \boldsymbol{w}),$$
$$(4)$$

where $\{\phi_j(x, \boldsymbol{w})\}_{j=1}^{\infty}$ is a set of infinite basis functions that satisfy the following conditions where Sobolev norm $W^{l,2}$ norm $(0 \le l \le \infty)$ is used:

**Assumption 4.1.** *The basis functions $\{\phi_j(x, \boldsymbol{w})\}_{j=1}^{\infty}$ are orthonormal basis functions, and satisfy $\|\phi_j(x, \boldsymbol{w})\|_{W^{l,2}} < \infty$ for all $j = 1, 2, \ldots$.*

**Assumption 4.2.** $\sum_{j=1}^{J} \beta_j \phi_j(x, \boldsymbol{w})$ *convergences uniformly to $g_0(x, \boldsymbol{w})$ if $J \to \infty$.*

We note that Hermite polynomial functions satisfy Assumption 4.2 for any bounded and continuous function $g_0$ [Damelin et al., 2001].

**Compactness Restriction.** The integral equation (3), known as a "Fredholm Integral Equation of the First Kind" [Bôcher, 1926], is ill-posed since the integral operator $\mathcal{K}$, where $\mathcal{K}(f)(z) = \int_{\Omega_{\boldsymbol{W}}} \int_{\Omega_X} k(z, x, \boldsymbol{w}) f(x, \boldsymbol{w}) dx d\boldsymbol{w}$, is not guaranteed to be compact. Problems where one or more of the three properties - existence, uniqueness, and stability of the solution - do not hold are called ill-posed problems [Tikhonov et al., 1995] and lead to severe estimation difficulties. To relieve the issue, we put restrictions on the functional space of $g_0(x, \boldsymbol{w})$. Let $\mathfrak{g}(X, \boldsymbol{W}) = \int_{\Omega_{\boldsymbol{W}}} \int_{\Omega_X} \{\mathbb{1}_{X \le x, \boldsymbol{W}=\boldsymbol{w}} - \mathbb{E}[\mathbb{1}_{X \le x, \boldsymbol{W}=\boldsymbol{w}} | Z = z_0]\} g(X, \boldsymbol{W}) dx d\boldsymbol{w}$, and define regularized Sobolev norm $\tilde{W}^{l,2}$, which is called "consistency norm" in [Gallant and Nychka, 1987], as follows

$$
\begin{aligned}
\|\mathfrak{g}(x, \boldsymbol{w})\|_{\tilde{W}^{l,2}}^2 = \sum_{|\lambda| \le l} \int & \left\{ D^\lambda \mathfrak{g}(x, \boldsymbol{w}) \right\}^2 \\
& \times \{1 + (x, \boldsymbol{w}^T)(x, \boldsymbol{w}^T)^T\}^\kappa dx d\boldsymbol{w},
\end{aligned} \tag{5}
$$

where $l \ge 1$ is an integer and $\kappa$ is a constant satisfying $\kappa > (1+d)/2$ where $d$ is the dimension of $\boldsymbol{W}$. We make the following assumption:

**Assumption 4.3.** *Given a positive regularization parameter $B_S$, $g_0(x, \boldsymbol{w})$ is in the functional space $\mathcal{G}_{B_S} = \{g : \|\mathfrak{g}(x, \boldsymbol{w})\|_{\tilde{W}^{l,2}}^2 \le B_S\}$.*

Using the approximation in (4), equation (3) reduces to

$$
\mu(z) = \sum_{j=1}^J \beta_j \int_{\Omega_{\boldsymbol{W}}} \int_{\Omega_X} k(z, x, \boldsymbol{w}) \phi_j(x, \boldsymbol{w}) dx d\boldsymbol{w}. \tag{6}
$$

Letting the anti-derivative of the basis functions be $\varphi_j(x, \boldsymbol{w}) = \int \phi_j(x, \boldsymbol{w}) dx$.[1] Then, the equation becomes

$$
\begin{aligned}
& \mathbb{E}[Y|Z=z] - \mathbb{E}[Y|Z=z_0] = \\
& \sum_{j=1}^J \beta_j \{\mathbb{E}[\varphi_j(X, \boldsymbol{W})|Z=z] - \mathbb{E}[\varphi_j(X, \boldsymbol{W})|Z=z_0]\}
\end{aligned} \tag{7}
$$

Let $c = \mathbb{E}[Y|Z=z] - \mathbb{E}[Y|z=z_0]$, $\boldsymbol{\beta} = (\beta_1, \ldots, \beta_J)^T$, and $\boldsymbol{d} = (d^1, \ldots, d^J)^T$ where $d^j = \mathbb{E}[\varphi_j(X, \boldsymbol{W})|Z=z] - \mathbb{E}[\varphi_j(X, \boldsymbol{W})|Z=z_0]$. Then, the integral equation (3) finally reduces to a linear equation $c = \boldsymbol{\beta}^T \boldsymbol{d}$.

**Sieve CAPCE (S-CAPCE) estimator.** Given datasets $\mathcal{D}^{(1)} = \{x_i^{(1)}, \boldsymbol{w}_i^{(1)}, z_i^{(1)}\}_{i=1}^{N_1}$ and $\mathcal{D}^{(2)} = \{y_i^{(2)}, z_i^{(2)}\}_{i=1}^{N_2}$, our S-CAPCE estimator consists of two stages. In Stage 1, we learn models $\hat{\mathbb{E}}[Y|Z=z]$ and $\hat{\mathbb{E}}[\varphi_j(X, \boldsymbol{W})|Z=z]$ from the datasets by regression. Then in Stage 2, we estimate parameters $\boldsymbol{\beta}$ by solving Eq. (7).

[1] We will simply write the antiderivative $\varphi_j(x, \boldsymbol{w}) = \int_{-\infty}^x \phi_j(x', \boldsymbol{w}) dx'$ as $\varphi_j(x, \boldsymbol{w}) = \int \phi_j(x, \boldsymbol{w}) dx$ in the paper.

**Stage 1.** We learn prediction models $\hat{\mathbb{E}}[Y|Z=z]$ using $\mathcal{D}^{(2)}$ and $\hat{\mathbb{E}}[\varphi_j(X, \boldsymbol{W})|Z=z]$ for $j = 1, \ldots, J$ using $\mathcal{D}^{(1)}$. Any regression method can be used. We select an IV value $z_0$. Denote $\hat{c}_i = \hat{\mathbb{E}}[Y|Z=z_i] - \hat{\mathbb{E}}[Y|Z=z_0]$ and $\hat{d}_i^j = \hat{\mathbb{E}}[\varphi_j(X, \boldsymbol{W})|Z=z_i] - \hat{\mathbb{E}}[\varphi_j(X, \boldsymbol{W})|Z=z_0]$.

Specifically, we perform the regression using the power series basis functions in this paper. Let basis functions be $\boldsymbol{q}(z) = (q_1(z), q_2(z), \ldots, q_P(z))^T$, and consider the model $\hat{\mathbb{E}}[Y|Z=z] = \sum_{p=1}^P \omega_p q_p(z)$, $\hat{\mathbb{E}}[\varphi_j(X, \boldsymbol{W})|Z=z] = \sum_{p=1}^P \nu_p^j q_p(z)$ for $j = 1, \ldots, J$. Denote $\boldsymbol{\omega} = (\omega_1, \ldots, \omega_P)^T$ and $\boldsymbol{\nu}^j = (\nu_1^j, \ldots, \nu_P^j)^T$. Then, we optimize the error functions below:

$$
\begin{aligned}
& Q_1(\boldsymbol{\nu}^j; \mathcal{D}^{(1)}) \\
& = \frac{1}{N_1} \sum_{i=1}^{N_1} (\varphi_j(x_i^{(1)}, \boldsymbol{w}_i^{(1)}) - \boldsymbol{q}(z_i^{(1)})^T \boldsymbol{\nu}^j)^2,
\end{aligned} \tag{8}
$$

$$
Q_2(\boldsymbol{\omega}; \mathcal{D}^{(2)}) = \frac{1}{N_2} \sum_{i=1}^{N_2} (y_i^{(2)} - \boldsymbol{q}(z_i^{(2)})^T \boldsymbol{\omega})^2. \tag{9}
$$

Let variance-covariance matrices be $\hat{\mathbf{M}}^{(1)} = \sum_{i=1}^{N_1} N_1^{-1} \boldsymbol{q}(z_i^{(1)}) \boldsymbol{q}(z_i^{(1)})^T$ and $\hat{\mathbf{M}}^{(2)} = \sum_{i=1}^{N_2} N_2^{-1} \boldsymbol{q}(z_i^{(2)}) \boldsymbol{q}(z_i^{(2)})^T$. We obtain

$$
\begin{cases}
\hat{c}_i = (\boldsymbol{q}(z_i) - \boldsymbol{q}(z_0))^T \hat{\mathbf{M}}^{(2)-} \sum_{l=1}^{N_2} \frac{1}{N_2} \boldsymbol{q}(z_l^{(2)}) y_l^{(2)} \\
\hat{d}_i^j = (\boldsymbol{q}(z_i) - \boldsymbol{q}(z_0))^T \hat{\mathbf{M}}^{(1)-} \\
\qquad \times \sum_{l=1}^{N_1} \frac{1}{N_1} \boldsymbol{q}(z_l^{(1)}) \varphi_j(x_l^{(1)}, \boldsymbol{w}_l^{(1)})
\end{cases} \tag{10}
$$

for $j = 1, \ldots, J$, where $\hat{\mathbf{M}}^-$ denotes the generalized inverse that satisfies $\hat{\mathbf{M}} \hat{\mathbf{M}}^- \hat{\mathbf{M}} = \hat{\mathbf{M}}$. Let $N = N_1 + N_2$ and $(z_1, \ldots, z_N) = (z_1^{(1)}, \ldots, z_{N_1}^{(1)}, z_1^{(2)}, \ldots, z_{N_2}^{(2)})$. We will compute predicted values in (10) for all $i = 1, \ldots, N$.

**Stage 2.** Estimate parameters $\boldsymbol{\beta}$ based on the linear equation $c = \boldsymbol{\beta}^T \boldsymbol{d}$. Let $\hat{\boldsymbol{c}} = (\hat{c}_1, \ldots, \hat{c}_N)^T$, $\hat{\boldsymbol{d}}_i = (\hat{d}_i^1, \ldots, \hat{d}_i^J)^T$, $\hat{\mathbf{D}} = (\hat{\boldsymbol{d}}_1, \ldots, \hat{\boldsymbol{d}}_N)^T$, and the empirical risk be

$$
Q_3(\boldsymbol{\beta}; \mathcal{D}^{(1)}, \mathcal{D}^{(2)}) = \frac{1}{N} \sum_{i=1}^N (\hat{c}_i - \hat{\boldsymbol{d}}_i^T \boldsymbol{\beta})^2. \tag{11}
$$

Under Assumption 4.3, our estimator $\hat{\boldsymbol{\beta}}$ is given by the optimization problem below:

$$
\min_{\boldsymbol{\beta}} Q_3(\boldsymbol{\beta}; \mathcal{D}^{(1)}, \mathcal{D}^{(2)}) \text{ subject to } \boldsymbol{\beta}^T \boldsymbol{\Lambda} \boldsymbol{\beta} \le B_S, \tag{12}
$$

where

$$
\begin{aligned}
\Lambda_{i,j} = \sum_{|\lambda| \le l} \int & \left\{ D^\lambda \varphi_i(x, \boldsymbol{w}) - D^\lambda \mathbb{E}[\varphi_i(X, \boldsymbol{W})|Z=z_0] \right\} \\
& \times \left\{ D^\lambda \varphi_j(x, \boldsymbol{w}) - D^\lambda \mathbb{E}[\varphi_j(X, \boldsymbol{W})|Z=z_0] \right\} \\
& \times \{1 + \|(x, \boldsymbol{w}^T)\|^2\}^\kappa dx d\boldsymbol{w}
\end{aligned} \tag{13}
$$

for $i, j = 1, \ldots, J$, and $\mathbf{\Lambda} = \{\Lambda_{i,j}\}_{i,j=1}^J$. $\mathbf{\Lambda}$ can be calculated by Monte Carlo integration $\hat{\mathbf{\Lambda}}$ [Kroese et al., 2011]. The optimization problem (12) can be solved by a ridge regression method with the following solution [Hilt et al.]:

$$\hat{\boldsymbol{\beta}} = (\hat{\mathbf{D}}^T \hat{\mathbf{D}} + \zeta_S \mathrm{diag}[\mathbf{\Lambda}])^{-1} \hat{\mathbf{D}}^T \hat{\boldsymbol{c}}, \qquad (14)$$

where $\zeta_S$ is a regularization parameter called Lagrange multipliers. Then, our proposed sieve CAPCE estimator is given by $\hat{\mathbb{E}}[\partial_x Y_x | \boldsymbol{w}] = \sum_{j=1}^J \hat{\beta}_j \phi_j(x, \boldsymbol{w})$.

**Model Selection.** The model selection in Stage 1 is a standard regression problem, and we presume the models in Stage 1 have been selected appropriately according to standard machine learning methods. We can use the empirical risk in equation (11) as a performance metric of the trained model in Stage 2 with parameters $\hat{\boldsymbol{\beta}}$ if given separate test datasets $\mathcal{D}^{(1)'} = \{x_i^{(1)'}, z_i^{(1)'}, \boldsymbol{w}_i^{(1)'}\}_{i=1}^{N_1'}$ and $\mathcal{D}^{(2)'} = \{y_i^{(2)'}, z_i^{(2)'}\}_{i=1}^{N_2'}$. Let $N' = N_1' + N_2'$. Assume $\hat{c}_i'$ and $\hat{\boldsymbol{d}}_i'$ for $i = 1, \ldots, N'$ are computed using $\mathcal{D}^{(1)'}$ and $\mathcal{D}^{(2)'}$. Then, we can evaluate the trained model by the test error $\hat{Q}_3(\hat{\boldsymbol{\beta}}; \mathcal{D}^{(1)'}, \mathcal{D}^{(2)'}) = \frac{1}{N'} \sum_{i=1}^{N'} (\hat{c}_i' - \hat{\boldsymbol{d}}_i'^T \hat{\boldsymbol{\beta}})^2$. Given separate datasets, this performance metric can be used for model selection from various candidate basis functions or the number $J$ or $P$ of basis terms.

**Property of sieve CAPCE estimator.** We show that sieve CAPCE estimator is consistent under assumptions similar to sieve NTSLS [Newey and Powell, 2003]. Assumptions B.1 - 4 are shown in Appendix B.

**Theorem 4.1** (Consistency). *Under SCM $\mathcal{M}_{IV}$ and Assumptions 3.1, 3.2, 4.1, 4.2, 4.3, B.1, B.2, B.3, and B.4, letting $P \to \infty$ and $J \to \infty$, then $\|\hat{g} - g_0\|_{W^{l,\infty}} \xrightarrow{p} 0$.*

**Theorem 4.2** (Rate of Convergence). *Under SCM $\mathcal{M}_{IV}$ and Assumptions 3.1, 3.2, 4.1, 4.2, 4.3, C.1, C.2, C.3, C.4, C.5, C.6, and C.7, setting $N = N_1 = N_2$, then $\|\hat{g} - g_0\|_A = o_p(N^{-1/4})$.*

Assumps. C.1-7 and norm $\|\cdot\|_A$ are defined in Appendix C.

## 4.2 PARAMETRIC CAPCE ESTIMATOR

Next, we develop a parametric CAPCE (P-CAPCE) estimator. We consider the setting that the CAPCE $\mathbb{E}[\partial_x Y_x | \boldsymbol{w}]$ takes the form of the following parametric model:

$$\mathbb{E}[\partial_x Y_x | \boldsymbol{w}] = \sum_{k=1}^K \gamma_k \theta_k(x, \boldsymbol{w}), \qquad (15)$$

where $\{\theta_k(x, \boldsymbol{w})\}_{k=1}^K$ are a set of known functions, and $\boldsymbol{\gamma} = (\gamma_1, \ldots, \gamma_K)^T$ are unknown model parameters to be estimated from data.

The derivation of the P-CAPCE estimator is very similar to that of the sieve CAPCE estimator, so we skip the details in the following. Denote the anti-derivatives $\vartheta_k(x, \boldsymbol{w}) = \int \theta_k(x, \boldsymbol{w}) dx$ for $k = 1, \ldots, K$. Let $c = \mathbb{E}[Y | Z = z] - \mathbb{E}[Y | z = z_0]$, and $\boldsymbol{e} = (e^1, \ldots, e^K)^T$ where $e^k = \mathbb{E}[\vartheta_k(X, \boldsymbol{W}) | Z = z] - \mathbb{E}[\vartheta_k(X, \boldsymbol{W}) | Z = z_0]$. Then, equation (3) reduces to a linear equation $c = \boldsymbol{\gamma}^T \boldsymbol{e}$.

**P-CAPCE estimator.** Given datasets $\mathcal{D}^{(1)} = \{x_i^{(1)}, \boldsymbol{w}_i^{(1)}, z_i^{(1)}\}_{i=1}^{N_1}$ and $\mathcal{D}^{(2)} = \{y_i^{(2)}, z_i^{(2)}\}_{i=1}^{N_2}$, our P-CAPCE estimator consists of two stages.

**Stage 1.** Let basis functions be $\boldsymbol{q}(z) = (q_1(z), q_2(z), \ldots, q_P(z))^T$. Denote $\hat{c}_i = \hat{\mathbb{E}}[Y | Z = z_i] - \hat{\mathbb{E}}[Y | Z = z_0]$ and $\hat{e}_i^k = \hat{\mathbb{E}}[\vartheta_k(X, \boldsymbol{W}) | Z = z_i] - \hat{\mathbb{E}}[\vartheta_k(X, \boldsymbol{W}) | Z = z_0]$. Let variance-covariance matrices $\hat{\mathbf{M}}^{(1)} = \sum_{i=1}^{N_1} N_1^{-1} \boldsymbol{q}(z_i^{(1)}) \boldsymbol{q}(z_i^{(1)})^T$ and $\hat{\mathbf{M}}^{(2)} = \sum_{i=1}^{N_2} N_2^{-1} \boldsymbol{q}(z_i^{(2)}) \boldsymbol{q}(z_i^{(2)})^T$. We obtain the following predication values

$$\begin{cases} \hat{c}_i = (\boldsymbol{q}(z_i) - \boldsymbol{q}(z_0))^T \hat{\mathbf{M}}^{(2)-} \sum_{l=1}^{N_2} \frac{1}{N_2} \boldsymbol{q}(z_l^{(2)}) y_l^{(2)} \\ \hat{e}_i^k = (\boldsymbol{q}(z_i) - \boldsymbol{q}(z_0))^T \hat{\mathbf{M}}^{(1)-} \\ \qquad \times \sum_{l=1}^{N_1} \frac{1}{N_1} \boldsymbol{q}(z_l^{(1)}) \vartheta_k(x_l^{(1)}, \boldsymbol{w}_l^{(1)}) \end{cases} \quad (16)$$

for $k = 1, \ldots, K$. Let $N = N_1 + N_2$ and $(z_1, \ldots, z_N) = (z_1^{(1)}, \ldots, z_{N_1}^{(1)}, z_1^{(2)}, \ldots, z_{N_2}^{(2)})$. We will compute predicted values in (16) for all $i = 1, \ldots, N$.

**Stage 2.** Estimate parameters $\boldsymbol{\gamma}$ based on the linear equation $c = \boldsymbol{\gamma}^T \boldsymbol{e}$. Let $\hat{\boldsymbol{c}} = (\hat{c}_1, \ldots, \hat{c}_N)^T$, $\hat{\boldsymbol{e}}_i = (\hat{e}_i^1, \ldots, \hat{e}_i^K)^T$, $\hat{\mathbf{E}} = (\hat{\boldsymbol{e}}_1, \ldots, \hat{\boldsymbol{e}}_N)^T$, and the empirical risk be

$$Q_4(\boldsymbol{\gamma}; \mathcal{D}^{(1)}, \mathcal{D}^{(2)}) = \sum_{i=1}^N \frac{1}{N} (\hat{c}_i - \hat{e}_i^T \boldsymbol{\gamma})^2. \qquad (17)$$

We make the following assumption:

**Assumption 4.4.** *Given a positive regularization parameter $B_P$, $\boldsymbol{\gamma}$ satisfies $\boldsymbol{\gamma}^T \boldsymbol{\gamma} \le B_P$.*

Under Assumption 4.4, our estimator $\hat{\boldsymbol{\gamma}}$ is given by the optimization problem below:

$$\min_{\boldsymbol{\gamma}} Q_4(\boldsymbol{\gamma}; \mathcal{D}^{(1)}, \mathcal{D}^{(2)}) \text{ subject to } \boldsymbol{\gamma}^T \boldsymbol{\gamma} \le B_P. \quad (18)$$

This problem can be solved by the ridge regression method with the following solution [Hilt et al.]:

$$\hat{\boldsymbol{\gamma}} = (\hat{\mathbf{E}}^T \hat{\mathbf{E}} + \zeta_P \mathbf{I}_K)^{-1} \hat{\mathbf{E}}^T \hat{\boldsymbol{c}}, \qquad (19)$$

where $\zeta_P$ is a regularization parameter, and $\mathbf{I}_K$ is a $K \times K$ identity matrix. Then, our proposed P-CAPCE estimator is given by $\hat{\mathbb{E}}[\partial_x Y_x | \boldsymbol{w}] = \sum_{k=1}^K \hat{\gamma}_k \theta_k(x, \boldsymbol{w})$.

**Model Selection.** We presume the models in Stage 1 have been selected appropriately. We can use the empirical risk in equation (17) as a performance metric of the

trained model in Stage 2 with parameters $\hat{\gamma}$ if given separate test datasets $\mathcal{D}^{(1)'} = \{x_i^{(1)'}, z_i^{(1)'}, \boldsymbol{w}_i^{(1)'}\}_{i=1}^{N_1'}$ and $\mathcal{D}^{(2)'} = \{y_i^{(2)'}, z_i^{(2)'}\}_{i=1}^{N_2'}$. Let $N' = N_1' + N_2'$. Assume $\hat{c}_i'$ and $\hat{e}_i'$ for $i = 1, \ldots, N$ are computed using $\mathcal{D}^{(1)'}$ and $\mathcal{D}^{(2)'}$. Then, we can evaluate the trained model by the test error $\hat{Q}_4(\hat{\gamma}\,; \mathcal{D}^{(1)'}, \mathcal{D}^{(2)'}) = \frac{1}{N'} \sum_{i=1}^{N'} (\hat{c}_i' - \hat{e}_i'^T \hat{\gamma})^2$. Given a separate dataset, this performance metric can be used for model selection from various candidate basis functions or the number $K$ or $P$ of basis terms.

**Property of P-CAPCE estimator.** We show that P-CAPCE estimator is consistent.

**Theorem 4.3** (Consistency). *Under SCM $\mathcal{M}_{IV}$ and Assumptions 3.1, 3.2, 4.4, D.1, D.2, D.3, and D.4, letting $P \to \infty$, then $\|\hat{\gamma} - \gamma\| \xrightarrow{p} 0$.*

**Theorem 4.4** (Rate of Convergence). *Under SCM $\mathcal{M}_{IV}$ and Assumptions 3.1, 3.2, 4.4, E.1, E.2, E.3, E.4, and E.5, setting $N = N_1 = N_2$, then $\|\hat{\gamma} - \gamma\| = o_p(N^{-1/4})$.*

Assumptions D.1 - 4 are shown in Appendix D. Assumptions E.1 - 5 are in Appendix E.

### 4.3 RKHS CAPCE ESTIMATOR

Finally, we develop a reproducing kernel Hilbert space (RKHS) CAPCE estimator. RKHS models are popular and widely used in nonparametric regression [Theodoridis and Koutroumbas, 2006, Schölkopf et al., 2013].

**RKHS model.** Let $k_{X,\boldsymbol{W}} : \Omega_{X,\boldsymbol{W}} \times \Omega_{X,\boldsymbol{W}} \to \mathbb{R}$ and $k_Z : \Omega_Z \times \Omega_Z \to \mathbb{R}$ be measurable positive definitive kernels corresponding to RKHSs $\mathcal{H}_{X,\boldsymbol{W}}$ and $\mathcal{H}_Z$. A symmetric function $k : \Omega \times \Omega \to \mathbb{R}$ is called positive-definite kernel if $\sum_{i=1}^{n} \sum_{j=1}^{n} c_i c_j k(\boldsymbol{a}_i, \boldsymbol{a}_j) \geq 0$ for all $\boldsymbol{a}_1, \ldots, \boldsymbol{a}_n \in \Omega$ given any $n \in \mathbb{N}$ and $c_1, \ldots, c_n \in \mathbb{R}$ [Shawe-Taylor and Cristianini, 2004]. Denote the feature map $\eta : \Omega_{X,\boldsymbol{W}} \to \mathcal{H}_{X,\boldsymbol{W}}, (x, \boldsymbol{w}) \mapsto k_{X,\boldsymbol{W}}'(x, \boldsymbol{w}, \cdot, \cdot)$ and $\psi : \Omega_Z \to \mathcal{H}_Z$, $z \mapsto k_Z(z, \cdot)$. In addition, we denote the antiderivative feature function $\pi : \Omega_{X,\boldsymbol{W}} \to \mathcal{H}_{X,\boldsymbol{W}}, (x, \boldsymbol{w}) \mapsto k_{X,\boldsymbol{W}}(x, \boldsymbol{w}, \cdot, \cdot)$ with $\pi(x, \boldsymbol{w}) = -\int \eta(x, \boldsymbol{w}) dx$ and the antiderivative kernel function $k_{X,\boldsymbol{W}}(x, \boldsymbol{w}, x', \boldsymbol{w}') = \int k_{X,\boldsymbol{W}}'(x, \boldsymbol{w}, x', \boldsymbol{w}') dx dx'$. Assume that the CAPCE takes the form

$$\mathbb{E}[\partial_x Y_x | \boldsymbol{w}] = H(\pi(x, \boldsymbol{w})) \quad (20)$$

for some operator $H \in \mathcal{L}_2(\mathcal{H}_{X,\boldsymbol{W}}, \Omega_Y)$, where $\mathcal{L}_2(\Omega_1, \Omega_2)$ is the $\mathcal{L}_2$ measurable function space from $\Omega_1$ to $\Omega_2$, and $H(\pi(x, \boldsymbol{w}))$ is a composition function $H \circ \pi : \Omega_{X,\boldsymbol{W}} \to$

$\Omega_Y$. Our RKHS CAPCE estimator consists of two stages (a detailed derivation is provided in Appendix A.2).

**Stage 1.** We learn an operator $G_1 \in \mathcal{L}_2(\mathcal{H}_Z, \mathcal{H}_{X,\boldsymbol{W}})$ that satisfies $\mathbb{E}[\pi(X, \boldsymbol{W})|Z = z] = G_1(\psi(z))$, and learn an operator $G_2 \in \mathcal{L}_2(\mathcal{H}_Z, \Omega_Y)$ that satisfies $\mathbb{E}[Y|Z = z] = G_2(\psi(z))$.

**Stage 2.** We learn an operator $H \in \mathcal{L}_2(\mathcal{H}_{X,\boldsymbol{W}}, \Omega_Y)$ that satisfies $\hat{\mathbb{E}}[Y|Z = z] - \hat{\mathbb{E}}[Y|Z = z_0] = H(\hat{\mathbb{E}}[\pi(X, \boldsymbol{W})|Z = z] - \hat{\mathbb{E}}[\pi(X, \boldsymbol{W})|Z = z_0]) \Leftrightarrow \hat{G}_1(\psi(z) - \psi(z_0)) = H(\hat{G}_2(\psi(z) - \psi(z_0)))$, where $\hat{G}_1$ and $\hat{G}_2$ are learned in Stage 1.

We learn $\hat{G}_1$, $\hat{G}_2$, and $\hat{H}$ by the following optimization problems using datasets $\mathcal{D}^{(1)}$ and $\mathcal{D}^{(2)}$:

$$\min_{G_1} \frac{1}{N_1} \sum_{i=1}^{N_1} \left\| \pi(x_i^{(1)}, \boldsymbol{w}_i^{(1)}) - G_1(\psi(z_i^{(1)})) \right\|_{\mathcal{H}_{X,\boldsymbol{W}}}^2 \quad (21)$$
$$+ \lambda_1 \|G_1\|_{\mathcal{L}_2(\mathcal{H}_Z, \mathcal{H}_{X,\boldsymbol{W}})}^2,$$

$$\min_{G_2} \frac{1}{N_2} \sum_{i=1}^{N_2} \left\| y_i^{(2)} - G_2(\psi(z_i^{(2)})) \right\|^2 \quad (22)$$
$$+ \lambda_2 \|G_2\|_{\mathcal{L}_2(\mathcal{H}_Z, \Omega_Y)}^2,$$

$$\min_{H} \frac{1}{N_2} \sum_{i=1}^{N_2} \left\| \hat{G}_2(\psi(z_i^{(2)}) - \psi(z_0)) \right.$$
$$\left. - H(\hat{G}_1(\psi(z_i^{(2)}) - \psi(z_0))) \right\|^2$$
$$+ \xi \|H\|_{\mathcal{L}_2(\mathcal{H}_{X,\boldsymbol{W}}, \Omega_Y)}^2 + \lambda_3 \left\| H \circ \hat{G}_1 \right\|_{\mathcal{L}_2(\mathcal{H}_Z, \Omega_Y)}^2, \quad (23)$$

where $(\lambda_1, \lambda_2, \lambda_3, \xi)$ are regularization parameters. From the representer theorem [Schölkopf et al., 2001], the optimal $G_1$ exists in span$\{\psi(z_1^{(1)}), \ldots, \psi(z_{N_1}^{(1)})\}$, and the optimal $G_2$ and $H$ exist in span$\{\psi(z_1^{(2)}), \ldots, \psi(z_{N_2}^{(2)})\}$.

We denote gram matrices $\mathbf{K}_{Z^{(1)}Z^{(1)}} = \{k_Z(z_i^{(1)}, z_j^{(1)})\}_{i,j=1}^{N_1}$; $\mathbf{K}_{Z^{(1)}z_0}$ is $N_1 \times N_1$ matrix $\{k_Z(z_i^{(1)}, z_0)\}_{i,j=1}^{N_1}$; and $\mathbf{K}_{(X,\boldsymbol{W})^{(1)}(x,\boldsymbol{w})}$ is $N_1$-dimension vector $\{k_{X,\boldsymbol{W}}(x_i^{(1)}, \boldsymbol{w}_i^{(1)}, x, \boldsymbol{w})\}_{i=1}^{N_1}$. Then, the RKHS CAPCE estimator is given by

$$\hat{\mathbb{E}}[\partial_x Y_x | \boldsymbol{w}] = \hat{\boldsymbol{\alpha}}^T \mathbf{K}_{(X,\boldsymbol{W})^{(1)}(x,\boldsymbol{w})}, \quad (24)$$

where

$$\hat{\boldsymbol{\alpha}} = (\hat{\mathbf{O}}\hat{\mathbf{O}}^T + N_2 \xi \mathbf{K}_{(X,\boldsymbol{W})^{(1)}(X,\boldsymbol{W})^{(1)}} + N_2 \lambda_3 \mathbf{I}_{N_2})^{-1}$$
$$\times \hat{\mathbf{O}}\{\boldsymbol{y}^{(2)T}(\mathbf{K}_{Z^{(2)}Z^{(2)}} + N_2 \lambda_2 \mathbf{I}_{N_2})^{-1}$$
$$\times (\mathbf{K}_{Z^{(2)}Z^{(2)}} - \mathbf{K}_{Z^{(2)}z_0})\}, \quad (25)$$

$$\hat{\mathbf{O}} = \mathbf{K}_{(X,\boldsymbol{W})^{(1)}(X,\boldsymbol{W})^{(1)}} (\mathbf{K}_{Z^{(1)}Z^{(1)}} + N_1\lambda_1\mathbf{I}_{N_1})^{-1}$$
$$\times (\mathbf{K}_{Z^{(1)}Z^{(2)}} - \mathbf{K}_{Z^{(1)}z_0}),$$
$$(26)$$

and $\mathbf{I}_N$ is a $N \times N$ identity matrix.

**Model Selection.** We presume the models in Stage 1 have been selected appropriately, and introduce a model selection method in Stage 2 following [Singh et al., 2019]. Assume we have separate datasets $\mathcal{D}^{(1)'} = \{x_i^{(1)'}, \boldsymbol{w}_i^{(1)'}, z_i^{(1)'}\}_{i=1}^{N_1'}$ and $\mathcal{D}^{(2)} = \{y_i^{(2)'}, z_i^{(2)'}\}_{i=1}^{N_2'}$. We determine the optimal $\lambda_1^*$ by minimizing

$$L_1(\lambda_1) = \frac{1}{N_1'}\text{Trace}\Big[\mathbf{K}_{(X,\boldsymbol{W})^{(1)'}(X,\boldsymbol{W})^{(1)'}}$$
$$- 2\mathbf{K}_{(X,\boldsymbol{W})^{(1)'}(X,\boldsymbol{W})^{(1)}}\mathbf{P}_1 + \mathbf{P}_1^T\mathbf{K}_{(X,\boldsymbol{W})^{(1)}(X,\boldsymbol{W})^{(1)}}\mathbf{P}_1\Big],$$
$$(27)$$

where $\mathbf{P}_1 = (\mathbf{K}_{Z^{(1)}Z^{(1)}} + N_1'\lambda_1\mathbf{I}_{N_1})^{-1}\mathbf{K}_{Z^{(1)}Z^{(2)}}$. We determine the optimal $\lambda_2^*$ by minimizing

$$L_2(\lambda_2) = \frac{1}{N_2'}\text{Trace}\Big[\boldsymbol{y}^{(2)'}\boldsymbol{y}^{(2)'T}$$
$$- 2\boldsymbol{y}^{(2)'}\boldsymbol{y}^{(2)T}\mathbf{P}_2 + \mathbf{P}_2^T\boldsymbol{y}^{(2)}\boldsymbol{y}^{(2)T}\mathbf{P}_2\Big],$$
$$(28)$$

where $\mathbf{P}_2 = (\mathbf{K}_{Z^{(1)}Z^{(1)}} + N_1\lambda_2\mathbf{I}_{N_1})^{-1}\mathbf{K}_{Z^{(1)}Z^{(2)}}$. Finally, we determine the optimal $\xi^*$ and $\lambda_3^*$ by minimizing test error $L(\lambda_3, \xi) = \frac{1}{N_2'}\sum_{i=1}^{N_2'}\|\boldsymbol{y}^{(2)'T}(\mathbf{K}_{Z^{(2)'}Z^{(2)'}} + N_2'\lambda_2^*\mathbf{I}_{N_2'})^{-1}(\mathbf{K}_{Z^{(2)'}Z^{(2)'}} - \mathbf{K}_{Z^{(2)'}z_0}) - \hat{H}_{\lambda_3,\xi}(x_i^{(1)'}, \boldsymbol{w}_i^{(1)'})\|^2$ where $\hat{H}_{\lambda_3,\xi}$ is learned with $\lambda_1 = \lambda_1^*$ and $\lambda_2 = \lambda_2^*$ using $\mathcal{D}^{(1)}$ and $\mathcal{D}^{(2)}$.

**Properties of RKHS CAPCE estimator.** The RKHS CAPCE estimator requires $\mathcal{O}(N_1^3) + \mathcal{O}(N_2^3)$ time [Saunders et al., 1998]. We show that RKHS CAPCE is consistent under assumptions similar to Kernel IV [Singh et al., 2019]. Assumptions F.1 - 8 are shown in Appendix F.

**Theorem 4.5** (Consistency). *Under SCM $\mathcal{M}_{IV}$ and Assumptions 3.1, 3.2, F.1, F.2, F.3, F.4, F.5, F.6, F.7 and F.8, the RKHS CAPCE estimator in (24) converges pointwise to CAPCE when $\lambda_3 = 0$.*

When $\lambda_3 = 0$, the inverse of the matrix $\hat{\mathbf{O}}\hat{\mathbf{O}}^T + N_2\xi\mathbf{K}_{(X,\boldsymbol{W})^{(1)}(X,\boldsymbol{W})^{(1)}}$ in Eq. (25) is numerically unstable. In practice, regularization leads to bias, but we must consider the bias-variance trade-off.

## 5 EXPERIMENTS

In this section, we present numerical experiments to demonstrate the performance of the proposed P-CAPCE, sieve CAPCE, and RKHS CAPCE estimators. Detailed settings are in Appendix G. The experiments are performed using an Apple M1 (16GB).

**Baselines.** We compare with the most widely used methods PTSLS (parametric), NTSLS (sieve), and Kernel IV. These methods compute $\mathbb{E}[Y_x|w]$ which we differentiate to compute CAPCE $\mathbb{E}[\partial_x Y_x|w]$.

**SCM Settings.** We consider the following two SCMs: $W := H + E_1, X := Z + W + H + E_2$, and

$$\begin{cases} Y := 10X^2 + WX + X + W + 50f(W)H + E_3 & \text{(A)} \\ Y := \exp(X)\exp(W) + 25f(W)H + E_3 & \text{(B)} \end{cases}$$
$$(29)$$

where $f(W) = W^5 + W^4 + W^3 + W^2$. The SCMs satisfy separability Assumption 3.2 but not (2). We use setting (A) as a parametric setting and setting (B) as a nonparametric setting. Values of $Z$, $H$, $E_1$, $E_2$, and $E_3$ are sampled i.i.d. from a uniform distribution on $[-1, 1]$. True CAPCE is $20x + w + 1$ in setting (A) and $\exp(x)\exp(w)$ in setting (B).

**Setting of P-CAPCE and PTSLS.** We used the basis terms $\{1, W, X\}$ for P-CAPCE and $\{1, W, X, WX, X^2\}$ for PTSLS, which match setting (A).

**Setting of NTSLS and sieve CAPCE.** We consider the basis terms $h_p(X)h_q(W)$ for $p = 0, 1, 2$ and $q = 0, 1, 2$, where $h_p$ is Hermite polynomial functions: $h_0(t) = 1$, $h_1(t) = t$, $h_2(t) = t^2 - 1$, and $h_3(t) = t^3 - 3t$.

**Setting of Kernel IV and RKHS CAPCE.** We use polynomial kernel function $k_Z(z, z') = (z^T z' + C_1)^{C_2}$ and $k_{X,W}(x, w, x', w') = ((x, w)^T(x', w') + C_3)^{C_4}$.

**Results.** The means of estimated coefficients by PTSLS and P-CAPCE in the parametric setting (A) are shown in Table 1. We observe that, when $N = 1000$, both P-CAPCE and PTSLS estimates have large standard deviations (SD) (shown in Appendix G) such that the differences in estimated values are not statistically significant. The estimated coefficients of P-CAPCE are converging to the true values when the sample size $N = 10000$, while the coefficient for $W$ estimated by PTSLS is still biased. We plotted the true and estimated CAPCE curves given $W = 1$ in Figure 2(a). It is clear that the estimated curve by P-CAPCE is much closer to the true curve than PTSLS. The true and estimated CAPCE surfaces over $(X, W)$ are shown in Appendix G.

We computed the mean-squared-error (MSE) between estimated and true CAPCE values for each estimator, where MSE is computed as $\frac{1}{N_1'}\sum_{i=1}^{N_1'}\{\hat{g}(x_i^{(1)'}, w_i^{(1)'}) - g(x_i^{(1)'}, w_i^{(1)'})\}^2$ with test dataset $\mathcal{D}^{(1)'}$, and the results are shown in Table 2. We observed that our sieve and RKHS CAPCE estimators are superior to the existing methods; sieve and RKHS CAPCE estimators are superior to P-CAPCE in the nonparametric setting (B); and kernel-based methods are much slower than other methods. We plotted the true and estimated CAPCE curves given $W = 1$ in Figure

Table 1: Means of estimated coefficients by PTSLS and P-CAPCE estimators in setting (A).

| Estimated coefficients | $N = 1000$ | | | $N = 10000$ | | |
|---|---|---|---|---|---|---|
| Terms | 1 | $W$ | $X$ | 1 | $W$ | $X$ |
| PTSLS | 1.248 | 50.032 | 27.862 | 1.101 | 51.181 | 19.763 |
| P-CAPCE | -1.651 | 10.383 | 19.293 | 1.226 | 0.963 | 19.971 |
| True Coefficients | 1 | 1 | 20 | 1 | 1 | 20 |

Table 2: MSE and run time of estimators in settings (A) and (B).

| MSE | PTSLS | NTSLS | Kernel IV | P-CAPCE | S-CAPCE | RKHS CAPCE |
|---|---|---|---|---|---|---|
| (A) $N = 1000$ | 925.139 | 418.396 | 548.821 | 104.990 | 203.079 | **87.853** |
| Time (second) | 0.126 | 0.361 | 6.105 | 0.132 | 0.596 | 6.410 |
| (A) $N = 10000$ | 817.074 | 357.777 | 495.742 | **69.185** | 185.056 | 71.276 |
| Time (second) | 0.372 | 1.127 | 2814.018 | 0.452 | 1.883 | 4530.765 |
| (B) $N = 1000$ | 290.340 | 46.405 | 45.734 | 202.313 | **8.600** | 11.612 |
| Time (second) | 0.127 | 0.356 | 6.019 | 0.143 | 0.454 | 6.540 |
| (B) $N = 10000$ | 265.400 | 20.990 | 51.470 | 54.124 | **3.579** | 8.985 |
| Time (second) | 0.367 | 1.031 | 2951.841 | 0.485 | 1.836 | 4360.991 |

2(b), which shows the estimated curves by sieve and RKHS CAPCE are much closer to the true curve than NTSLS and Kernel IV. The true and estimated CAPCE surfaces over $(X, W)$ are shown in Appendix G.

Overall, the results of settings (A) and (B) show that our proposed methods (P-CAPCE, sieve CAPCE, RKHS CAPCE) are superior to the previous works (PTSLS, NTSLS, Kernel IV). The advantage of our proposed methods stems from that the underlying models (A) and (B) do not satisfy the separability assumption (2) needed by the existing works. Indeed, we have performed experiments in settings where the interaction between the covariates $W$ and unobserved confounders $H$ (the $f(W)H$ term in (29)) is absent, and the results (presented in Appendix G) show that the performances of the existing methods PTSLS, NTSLS, Kernel IV are comparable with our proposed methods under this situation. Among the three proposed methods, the performance of P-CAPCE relies on correct parametric model assumption, and RKHS CAPCE is computationally expensive and requires tuning many regularization parameters.

## 6 APPLICATION IN A REAL-WORLD DATASET

In this section, we present an application of our CAPCE estimators to a real-world dataset in economics.

**Real-world Dataset.** We take up an open dataset "the National Longitudinal Survey of Young Men" in the R package "wooldridge" (https://cran.r-project.org/package=wooldridge), which has been analyzed by

many works, e.g., in [Griliches, 1977, Blackburn and Neumark, 1992]. The sample size is 935 with 857 left after excluding missing values. We evaluate the heterogeneity of the effect of years of education on monthly wages, which is of great interest in economics [Angrist and Krueger, 1991, Card, 1999]. We followed Blackburn and Neumark [1992] to use mother's education as an instrument to uncover the effect of education on wages. The use of mother's education as an instrument in this dataset has been subjected to debate in the literature (e.g., [Card, 1999, Kling, 2001, Wooldridge, 2010]). We take the subject's years of education as the treatment variable ($X$), their monthly wage as the outcome ($Y$), their mother's years of education as the IV ($Z$), and their IQ as a covariate ($W$). The domains of $X$ and $Z$ are $[9, 18]$, ranging from the 1st year of high school to the 2nd year of a master's degree. The domain of $W$ is $[50, 145]$.

**Settings.** We applied P-CAPCE and PTSLS. Other estimators are not used due to the small sample size. We use terms $\{1, W, W^2, X, XW, XW^2\}$ for P-CAPCE and $\{1, W, W^2, X, XW, XW^2, X^2, X^2W, X^2W^2\}$ for PTSLS. Detailed settings are in Appendix G.

**Results.** The estimated CAPCE values are shown in Appendix G. For subjects with IQ 100, the estimated CAPCE $\mathbb{E}[\partial_x Y_x | W = 100]$ of years of education ($X$) on wages ($Y$) is given by $94.905 - 5.618x$ by P-CAPCE and $108.491 - 5.882x$ by PTSLS. Both predict that years of education increase wages, which is consistent with previous works [Blackburn and Neumark, 1992, Wooldridge, 2010]. The results also show that education significantly affects wages at the compulsory school level, but the effect gets weaker with more years of education, consistent with the results

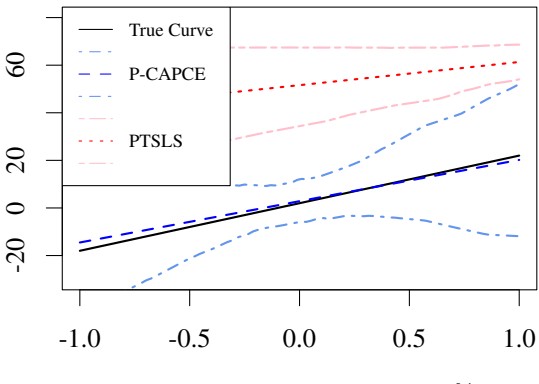

(a) Parametric setting (A) (Means, 95% CI)

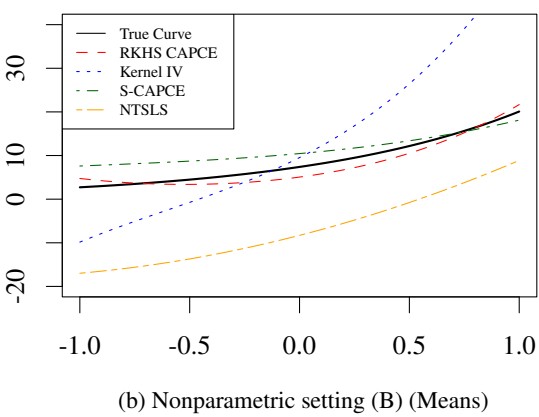

(b) Nonparametric setting (B) (Means)

Figure 2: Plots of CAPCE curves with $W = 1$. X-axis represents treatment ($X$); Y-axis is CAPCE value. Dot-dashed curves in (a) represent $95\%$ pointwise confidence interval (CI).

in [Angrist and Krueger, 1991, Caplan, 2018]. On the other hand, for subjects with IQ 80, the estimated CAPCE $\mathbb{E}[\partial_x Y_x | W = 80]$ is $60.740 - 3.598x$ by P-CAPCE and $69.465 - 3.057x$ by PTSLS. For subjects with IQ 120, the estimated CAPCE $\mathbb{E}[\partial_x Y_x | W = 120]$ is $136.662 - 8.086x$ by P-CAPCE and $156.181 - 9.531x$ by PTSLS.

While we estimate the heterogeneity of causal effects of education on wages across subjects with different IQs, existing works [Blackburn and Neumark, 1992, Card, 1999, Kling, 2001, Wooldridge, 2010, Kawakami et al., 2023] using this dataset have focused on the effects of education on wages over the whole population. Card [1999] and Wooldridge [2010] provided a summary of the early works on IV estimates and showed that the estimates of all studies were positive implying education increases wages. On the other hand, our results give two new insights into the effects of education on wages. First, our results suggest that for each sub-population IQ = 80, 100, 120, education significantly affects wages at the compulsory school level; but has little effect at the college level. This result is consistent with the

result of APCE estimates for the whole population given in [Kawakami et al., 2023]. Second, we reveal that the effect of education on wages is more significant for high IQ students, especially at the compulsory school level. To the best of our knowledge, this result has not been revealed in previous studies of this dataset, but it is consistent with the panel data analysis result in [Altonji and Dunn, 1996].

# 7 CONCLUSION

We study conditional average partial causal effect (CAPCE) to represent the heterogeneous causal effects of a continuous treatment. We present a method for identifying CAPCE in the IV model. Notably, CAPCE $\mathbb{E}[\partial_x Y_x | \boldsymbol{w}]$ is identifiable under a weaker assumption than required by $\mathbb{E}[Y_x | \boldsymbol{w}]$, showing the merit of studying CAPCE instead of $\mathbb{E}[Y_x | \boldsymbol{w}]$, which has been the focus of existing work. We develop three families of CAPCE estimators: sieve, parametric, and RKHS, and analyze their statistical properties. We empirically demonstrate that the proposed CAPCE estimators are superior to the existing widely used IV methods PT-SLS [Angrist and Pischke, 2009, Wooldridge, 2010], sieve NTSLS [Newey and Powell, 2003], and Kernel IV [Singh et al., 2019] in settings where the standard separability assumption (2) is violated. The work provides scientists with a new tool for analyzing the heterogeneous causal effects of a continuous treatment. The results can be extended to an IV model with an additional edge $\boldsymbol{W} \to Z$. An identification theorem similar to Theorem 3.1 can be derived, which uses $\mathbb{P}(Y | Z, \boldsymbol{W})$ as input instead of $\mathbb{P}(Y | Z)$. We present this result in Appendix A.2.

## ACKNOWLEDGEMENTS

The authors thank the anonymous reviewers for their time and thoughtful comments. Yuta Kawakami was supported by JSPS KAKENHI Grant Number 22J21928. Manabu Kuroki was supported by JSPS KAKENHI Grant Number 21H03504 and 24K14851. Jin Tian was partially supported by NSF grant CNS-2321786.

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

# Appendix to "Identification and Estimation of Conditional Average Partial Causal Effects via Instrumental Variable"

**Yuta Kawakami**[1,2]      **Manabu Kuroki**[1]      **Jin Tian**[2]

[1]Department of Mathematics, Physics, Electrical Engineering and Computer Science, Yokohama National University, Yokohama, Kanagawa, JAPAN
[2]Department of Computer Science, Iowa State University, Ames, Iowa, USA

## A    PROOFS OF THEOREM 3.1 AND RKHS CAPCE ESTIMATOR

### A.1    PROOF OF THEOREM 3.1

We give proof of Theorem 3.1.

**Theorem 3.1.** *(Identification of CAPCE). Under SCM $\mathcal{M}_{IV}$ and Assumptions 3.1 and 3.2, CAPCE $\mathbb{E}[\partial_x Y_x | \boldsymbol{w}]$ is identifiable from distributions $\mathbb{P}(X, \boldsymbol{W} | Z)$ and $\mathbb{P}(Y | Z)$ via the integral equation:*

$$\mu(z) = \int_{\Omega_{\boldsymbol{W}}} \int_{\Omega_X} k(z, x, \boldsymbol{w}) \mathbb{E}[\partial_x Y_x | \boldsymbol{w}] dx d\boldsymbol{w}, \tag{30}$$

*where $\mu(z) = \mathbb{E}[Y | Z = z_0] - \mathbb{E}[Y | Z = z], k(z, x, \boldsymbol{w}) = \mathfrak{p}(X \le x, \boldsymbol{W} = \boldsymbol{w} | Z = z) - \mathfrak{p}(X \le x, \boldsymbol{W} = \boldsymbol{w} | Z = z_0)$, and $z_0$ is a fixed value.*

*Proof.* First, we show the following integral equation holds under Assumptions 3.1 and 3.2 following the idea in [Wong, 2022]:

$$\mathbb{E}[Y | Z = z, \boldsymbol{W} = \boldsymbol{w}] - \mathbb{E}[Y | Z = z_0, \boldsymbol{W} = \boldsymbol{w}] \tag{31}$$
$$= -\int_{\Omega_X} \{\mathbb{P}(X \le x | Z = z, \boldsymbol{W} = \boldsymbol{w}) - \mathbb{P}(X \le x | Z = z_0, \boldsymbol{W} = \boldsymbol{w})\} \mathbb{E}[\partial_x Y_x | \boldsymbol{w}] dx.$$

From the setting of the IV, the following integral equation holds:

$$Y_{X_z} = \int_{\Omega_X} \mathbb{1}_{X_z = x} Y_x dx, \tag{32}$$

given $\boldsymbol{W} = \boldsymbol{w}$ for each subject, where $\mathbb{1}$ is a delta function or indicator function. This equation means $X_z = x \Rightarrow Y_{X_z} = Y_x$ from the definition of delta function. By substituting the integral equations $Y_{X_z} = \int_{\Omega_X} \mathbb{1}_{X_z = x} Y_x dx$ and $Y_{X_{z_0}} = \int_{\Omega_X} \mathbb{1}_{X_{z_0} = x} Y_x dx$, then

$$Y_{X_z} - Y_{X_{z_0}} = \int_{\Omega_X} \{\mathbb{1}_{X_z = x} - \mathbb{1}_{X_{z_0} = x}\} Y_x dx \tag{33}$$

holds. Since the Heaviside step function is the integration of the delta function,

$$Y_{X_z} - Y_{X_{z_0}} = \left[\{\mathbb{1}_{X_z = x} - \mathbb{1}_{X_{z_0} = x}\} \partial_x Y_x \right]_{-\infty}^{\infty} - \int_{\Omega_X} \{\mathbb{I}_{X_z \le x} - \mathbb{I}_{X_{z_0} \le x}\} \partial_x Y_x dx. \tag{34}$$

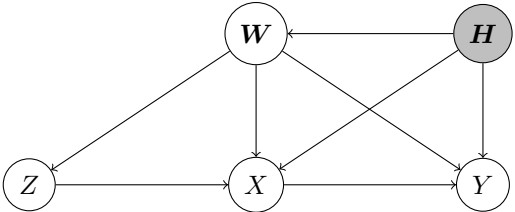

Figure 3: A causal graph representing the IV setting with covariates when there is an edge $\boldsymbol{W} \to Z$.

Because $\partial_x Y_x < \infty$ for all $x \in \Omega_X$, $\left[\{\mathbb{1}_{X_z=x} - \mathbb{1}_{X_{z_0}=x}\}\partial_x Y_x\right]_{-\infty}^{\infty} = 0$ holds. Then, the integral equation becomes

$$Y_{X_z} - Y_{X_{z_0}} = -\int_{\Omega_X} \{\mathbb{I}_{X_z \leq x} - \mathbb{I}_{X_{z_0} \leq x}\}\partial_x Y_x dx. \tag{35}$$

From the separability with covariate $f_Y(X, \boldsymbol{W}, \boldsymbol{H}, \boldsymbol{u}_Y) = f_Y^1(X, \boldsymbol{W}, \boldsymbol{u}_Y) + f_Y^2(\boldsymbol{W}, \boldsymbol{H}, \boldsymbol{u}_Y)$, random variables $\mathbb{I}_{X_z \leq x} - \mathbb{I}_{X_{z_0} \leq x}$ and $\partial_x Y_x$ are independent given $\boldsymbol{W} = \boldsymbol{w}$. Thus, we take expectations on both sides:

$$\mathbb{E}[Y_{X_z}|\boldsymbol{W} = \boldsymbol{w}] - \mathbb{E}[Y_{X_{z_0}}|\boldsymbol{W} = \boldsymbol{w}] \tag{36}$$

$$= -\int_{\Omega_X} \mathbb{E}[\{\mathbb{I}_{X_z \leq x} - \mathbb{I}_{X_{z_0} \leq x}\}\partial_x Y_x|\boldsymbol{W} = \boldsymbol{w}]dx \tag{37}$$

$$= -\int_{\Omega_X} \{\mathbb{E}[\mathbb{I}_{X_z \leq x}|\boldsymbol{W} = \boldsymbol{w}] - \mathbb{E}[\mathbb{I}_{X_{z_0} \leq x}|\boldsymbol{W} = \boldsymbol{w}]\}\mathbb{E}[\partial_x Y_x|\boldsymbol{w}]dx. \tag{38}$$

Then, the integral equation becomes

$$\mathbb{E}[Y|Z = z, \boldsymbol{W} = \boldsymbol{w}] - \mathbb{E}[Y|Z = z_0, \boldsymbol{W} = \boldsymbol{w}] \tag{39}$$

$$= -\int_{\Omega_X} \{\mathbb{P}(X \leq x|Z = z, \boldsymbol{W} = \boldsymbol{w}) - \mathbb{P}(X \leq x|Z = z_0, \boldsymbol{W} = \boldsymbol{w})\}\mathbb{E}[\partial_x Y_x|\boldsymbol{w}]dx.$$

Next, the integral equation can be given by multiplying $\mathfrak{p}(\boldsymbol{W} = \boldsymbol{w}|Z = z)$ and marginalizing for $\boldsymbol{W}$, then

$$\mathbb{E}_{\boldsymbol{W}}[\mathbb{E}[Y|Z = z, \boldsymbol{W} = \boldsymbol{w}]] = \tag{40}$$

$$\int_{\Omega_X} \int_{\Omega_{\boldsymbol{W}}} \mathbb{P}(X \leq x|Z = z, \boldsymbol{W} = \boldsymbol{w})\mathfrak{p}(\boldsymbol{W} = \boldsymbol{w}|Z = z)\mathbb{E}[\partial_x Y_x|\boldsymbol{w}]d\boldsymbol{w}dx \tag{41}$$

$$\Leftrightarrow \quad \mathbb{E}[Y|Z = z] = \int_{\Omega_X} \int_{\Omega_{\boldsymbol{W}}} \mathfrak{p}(X \leq x, \boldsymbol{W} = \boldsymbol{w}|Z = z)\mathbb{E}[\partial_x Y_x|\boldsymbol{w}]d\boldsymbol{w}dx \tag{42}$$

Finally, we show the uniqueness of the solution. Since $X_z$ is a nontrivial function, there does not exist a function which satisfies $\mathbb{E}[\delta(X)|Z = z, \boldsymbol{W} = \boldsymbol{w}] = 0$ for any $z \in \Omega_Z$ and $\boldsymbol{w} \in \Omega_{\boldsymbol{W}}$. Since $\mathbb{E}[\delta(X)|Z = z, \boldsymbol{W} = \boldsymbol{w}] = \mathbb{E}[\delta(X), \boldsymbol{W} = \boldsymbol{w}|Z = z]\mathbb{P}(\boldsymbol{W} = \boldsymbol{w})$, there exists a function which satisfies $\mathbb{E}[\delta(X), \boldsymbol{W} = \boldsymbol{w}|Z = z] = 0$ for any $z \in \Omega_Z$ and $\boldsymbol{w} \in \Omega_{\boldsymbol{W}}$ if there exists a function which satisfies $\mathbb{E}[\delta(X)|Z = z] = 0$ for any $z \in \Omega_Z$ and $\boldsymbol{w} \in \Omega_{\boldsymbol{W}}$. Taking a contraposition, there does not exist a function which satisfies $\mathbb{E}[\delta(X), \boldsymbol{W} = \boldsymbol{w}|Z = z] = 0$ for any $z \in \Omega_Z$ and $\boldsymbol{w} \in \Omega_{\boldsymbol{W}}$. $\qquad\square$

## A.2 IDENTIFICATION THEOREM UNDER IV MODEL IN FIG 3

We consider the IV model with covariates represented by the causal graph in Fig 3, with the following SCM $\mathcal{M}'_{IV}$ over $\boldsymbol{V} = \{Z, X, Y, \boldsymbol{W}\}$ and $\boldsymbol{U} = \{\boldsymbol{H}, \boldsymbol{u}_X, \boldsymbol{u}_Y, \boldsymbol{u}_Z, \boldsymbol{u}_{\boldsymbol{W}}\}$:

$$Y := f_Y(X, \boldsymbol{W}, \boldsymbol{H}, \boldsymbol{u}_Y), \ X := f_X(Z, \boldsymbol{W}, \boldsymbol{H}, \boldsymbol{u}_X), \boldsymbol{W} := f_{\boldsymbol{W}}(\boldsymbol{H}, \boldsymbol{u}_{\boldsymbol{W}}), \ Z := f_Z(\boldsymbol{W}, \boldsymbol{u}_Z), \tag{43}$$

where $f_{\boldsymbol{W}}$ is a vector function. We assume all variables are continuous, $\boldsymbol{W}$ are $d$-dimensional pre-treatment covariates, and $\boldsymbol{H}$ stands for unmeasured confounders. We show a similar identification result to Theorem 3.1.

**Theorem 3.1'.** *Under SCM $\mathcal{M}'_{IV}$ and Assumptions 3.1 and 3.2, CAPCE $\mathbb{E}[\partial_x Y_x | \boldsymbol{w}]$ is identifiable from distributions $\mathbb{P}(X|Z, \boldsymbol{W})$ and $\mathbb{P}(Y|Z, \boldsymbol{W})$ via the integral equation:*

$$\mu(z, \boldsymbol{w}) = \int_{\Omega_X} k(z, x, \boldsymbol{w}) \mathbb{E}[\partial_x Y_x | \boldsymbol{w}] dx, \tag{44}$$

*where $\mu(z, \boldsymbol{w}) = \mathbb{E}[Y|Z = z_0, \boldsymbol{W} = \boldsymbol{w}] - \mathbb{E}[Y|Z = z, \boldsymbol{W} = \boldsymbol{w}], k(z, x, \boldsymbol{w}) = \mathfrak{p}(X \le x | Z = z, \boldsymbol{W} = \boldsymbol{w}) - \mathfrak{p}(X \le x | Z = z_0, \boldsymbol{W} = \boldsymbol{w})$, and $z_0$ is a fixed value.*

*Proof.* Eq. (44) is guaranteed by Eq. (39), which appears in the proof of Theorem 3.1. $\square$

Based on Theorem 3.1', we have to learn $\mathbb{E}[\partial_x Y_x | \boldsymbol{w}]$ as a function of $x$ for each $\boldsymbol{w} \in \Omega_{\boldsymbol{W}}$ respectively. In contrast, based on Theorem 3.1, we can learn $\mathbb{E}[\partial_x Y_x | \boldsymbol{w}]$ directly as a function of $x$ and $\boldsymbol{w}$.

We perform experiments about estimating CAPCE based on Theorem 3.1' in Appendix G.5.

## A.3 DERIVATION OF RKHS CAPCE ESTIMATOR

We show the detailed steps of deriving the RKHS CAPCE estimator.

**RKHS estimator.** RKHS CAPCE estimator is given as $\hat{\mathbb{E}}[\partial_x Y_x | \boldsymbol{w}] = \hat{\boldsymbol{\alpha}}^T \mathbf{K}_{(X, \boldsymbol{W})^{(1)}}(x, \boldsymbol{w})$ where

$$\begin{aligned}
\hat{\boldsymbol{\alpha}} &= (\hat{\mathbf{O}}\hat{\mathbf{O}}^T + N_2 \xi \mathbf{K}_{(X, \boldsymbol{W})^{(1)}(X, \boldsymbol{W})^{(1)}} + N_2 \lambda_3 \mathbf{I}_{N_2})^{-1} \hat{\mathbf{O}} \\
&\quad \times \{\boldsymbol{y}^{(2)T} (\mathbf{K}_{Z^{(2)}Z^{(2)}} + N_2 \lambda_2 \mathbf{I}_{N_2})^{-1} (\mathbf{K}_{Z^{(2)}Z^{(2)}} - \mathbf{K}_{Z^{(2)}z_0})\} \\
\hat{\mathbf{O}} &= \mathbf{K}_{(X, \boldsymbol{W})^{(1)}(X, \boldsymbol{W})^{(1)}} (\mathbf{K}_{Z^{(1)}Z^{(1)}} + N_1 \lambda_1 \mathbf{I}_{N_1})^{-1} (\mathbf{K}_{Z^{(1)}Z^{(2)}} - \mathbf{K}_{Z^{(1)}z_0}),
\end{aligned} \tag{45}$$

$(\lambda_1, \lambda_2, \lambda_3, \xi)$ are regularization parameters, and $\mathbf{I}_N$ is a $N \times N$ identity matrix.

*Proof.* There are three optimization problems in RKHS estimator, **Stage 1 (A)** learning linear operator $G_1$, **Stage 1 (B)** learning linear operator $G_2$, and **Stage 2** learning linear operator $H$. We explain them respectively.

**Stage 1 (A).** We denote the feature map be $\psi(z)$ and $\pi(x, \boldsymbol{w})$, where $\pi(x, \boldsymbol{w}) = -\int_{-\infty}^{x} \eta(x', \boldsymbol{w}) dx'$ for some feature function $\eta(x', \boldsymbol{w})$. The optimization problem in **Stage 1 (A)** becomes

$$\min_{G_1 \in \mathcal{L}_2(\mathcal{H}_Z, \mathcal{H}_{X, \boldsymbol{W}})} N_1^{-1} \sum_{i=1}^{N_1} \left\| \pi(x_i^{(1)}, \boldsymbol{w}_i^{(1)}) - G_1(\psi(z_i^{(1)})) \right\|_{\mathcal{H}_{X, \boldsymbol{W}}}^2 + \lambda_1 \|G_1\|_{\mathcal{L}_2(\mathcal{H}_Z, \mathcal{H}_{X, \boldsymbol{W}})}^2. \tag{46}$$

using $\mathcal{D}^{(1)}$. Then, the estimator $\hat{G}_1$ becomes

$$\hat{G}_1(\cdot) = \left\langle \pi_{X^{(1)}, \boldsymbol{W}^{(1)}} (\mathbf{K}_{Z^{(1)}Z^{(1)}} + N_1 \lambda_1 \mathbf{I})^{-1} \psi_Z^{(1)T}, \cdot \right\rangle \tag{47}$$

where $\mathbf{K}_{Z^{(1)}Z^{(1)}}$ and $\mathbf{K}_{X^{(1)}X^{(1)}}$ are the empirical kernel matrices, the $i$-th column of $\pi_{X^{(1)}, \boldsymbol{W}^{(1)}}$ is $-\int_{-\infty}^{x_i^{(1)}} \eta(x, \boldsymbol{w}) dx$, and the $i$-th column of $\psi_X^{(1)}$ is $\psi(z_i^{(1)})$. The prediction values are

$$d_0(z) = \pi_{X^{(1)}, \boldsymbol{W}^{(1)}} (\mathbf{K}_{ZZ} + N_1 \lambda_1 I)^{-1} \psi_Z^{(1)T} \psi(z) = -\sum_{i=1}^{N_1} \gamma_i(z) \int_{-\infty}^{x_i^{(1)}} \eta(x, \boldsymbol{w}) dx \tag{48}$$

where $\gamma(z) = (\mathbf{K}_{Z^{(1)}Z^{(1)}} + N_1 \lambda_1 \mathbf{I})^{-1} \psi_Z^{(1)T} \psi(z) = (\mathbf{K}_{Z^{(1)}Z^{(1)}} + N_1 \lambda_1 \mathbf{I})^{-1} \mathbf{K}_{Z^{(1)}z}$. Furthermore, the difference in the predication values are

$$d(z) = d_0(z) - d_0(z_0) = -\sum_{i=1}^{N_1} \{\gamma_i(z) - \gamma_i(z_0)\} \int_{-\infty}^{x_i^{(1)}} \eta(x, \boldsymbol{w}) dx \tag{49}$$

and $\gamma(z) - \gamma_i(z_0) = (\mathbf{K}_{Z^{(1)}Z^{(1)}} + N_1\lambda_1\mathbf{I})^{-1}\psi_Z^{(1)T}\{\psi(z) - \psi(z_0)\} = (\mathbf{K}_{Z^{(1)}Z^{(1)}} + N_1\lambda_1\mathbf{I})^{-1}(\mathbf{K}_{Z^{(1)}z} - \mathbf{K}_{Z^{(1)}z_0})$ holds.
Letting $\hat{G}_1 = \sum_{j=1}^{N_1} \alpha_j\eta(x_j^{(1)}, \boldsymbol{w}_j^{(1)})$ since the optimal $\hat{G}_1$ exists in $\mathrm{span}(\{\eta(x_j^{(1)}, \boldsymbol{w}_j^{(1)})\}_{j=1}^{N_1})$ from the representer theorem
[Schölkopf et al., 2001]. Then the functional form of $d(z)$ is restricted by

$$d(z) = -\left\langle \sum_{j=1}^{N_1} \alpha_j \int_{-\infty}^{x_i^{(1)}} \eta(x, \boldsymbol{w}_i^{(1)})dx, -\sum_{i=1}^{N_1}\{\gamma_i(z) - \gamma_i(z_0)\}\int_{-\infty}^{x_j^{(1)}} \eta(x, \boldsymbol{w}_j^{(1)})dx \right\rangle \tag{50}$$

$$= \sum_{i=1}^{N_1}\sum_{j=1}^{N_1}\alpha_j\{\gamma_i(z) - \gamma_i(z_0)\}\left\langle -\int_{-\infty}^{x_i^{(1)}}\eta(x, \boldsymbol{w}_i^{(1)})dx, -\int_{-\infty}^{x_j^{(1)}}\eta(x, \boldsymbol{w}_j^{(1)})dx\right\rangle \tag{51}$$

$$= \sum_{i=1}^{N_1}\sum_{j=1}^{N_1}\alpha_j\{\gamma_i(z) - \gamma_i(z_0)\}\left\langle \pi(x_i^{(1)}, \boldsymbol{w}_i^{(1)}), \pi(x_j^{(1)}, \boldsymbol{w}_j^{(1)})\right\rangle. \tag{52}$$

From the kernel trick, it becomes

$$= \sum_{i=1}^{N_1}\sum_{j=1}^{N_1}\alpha_j\{\gamma_i(z) - \gamma_i(z_0)\}k((x_i^{(1)}, \boldsymbol{w}_i^{(1)}), (x_j^{(1)}, \boldsymbol{w}_j^{(1)})) \tag{53}$$

$$= \boldsymbol{\alpha}^T w(z) \tag{54}$$

where $w(z) = \mathbf{K}_{(X,\boldsymbol{W})^{(1)}(X,\boldsymbol{W})^{(1)}}(\mathbf{K}_{ZZ} + N_1\lambda_1\mathbf{I})^{-1}(\mathbf{K}_{Z^{(1)}z} - \mathbf{K}_{Z^{(1)}z_0})$. Note that the $\boldsymbol{\alpha}$ will be estimated in **Stage 2.**

**Stage 1 (B).** The optimization problem in **Stage 1 (B)** is

$$\min_{G_2 \in \mathcal{L}_2(\mathcal{H}_Z, \Omega_Y)} N_2^{-1}\sum_{i=1}^{N_2}\left\| y_i^{(2)} - G_2(\psi(z_i^{(2)}))\right\|^2 + \lambda_2\|G_2\|_{\mathcal{L}_2(\mathcal{H}_Z, \Omega_Y)}^2 \tag{55}$$

using $\mathcal{D}^{(2)}$. As **Stage 1 (A)**, the estimator of $G_2$, $\hat{G}_2$, become

$$\hat{G}_2(\cdot) = \left\langle \boldsymbol{y}^{(2)}(\mathbf{K}_{Z^{(2)}Z^{(2)}} + N_2\lambda_2\mathbf{I})^{-1}\psi_Z^{(2)T}, \cdot\right\rangle \tag{56}$$

where $\mathbf{K}_{Z^{(2)}Z^{(2)}}$ are the gram matrices, the $i$-th column of $\boldsymbol{y}^{(2)}$ is $y_i^{(2)}$.

$$u_0(z) = \boldsymbol{y}^{(2)T}(\mathbf{K}_{Z^{(2)}Z^{(2)}} + N_2\lambda_2\mathbf{I})^{-1}\psi_Z^{(2)T}\psi(z) = \sum_{i=1}^{N_2}\gamma_i(z)\psi(z) \tag{57}$$

where $\gamma(z) = \boldsymbol{y}^{(2)T}(\mathbf{K}_{Z^{(2)}Z^{(2)}} + N_2\lambda_2 I)^{-1}\psi_Z^{(2)T}$. Then,

$$u_0(z_0) = \boldsymbol{y}^{(2)T}(\mathbf{K}_{Z^{(2)}Z^{(2)}} + N_2\lambda_2\mathbf{I})^{-1}\mathbf{K}_{Z^{(2)}z_0} \tag{58}$$

and, the difference of the predication values are

$$u(z) = u_0(z) - u_0(z_0) = \boldsymbol{y}^{(2)T}(\mathbf{K}_{Z^{(2)}Z^{(2)}} + N_2\lambda_2\mathbf{I})^{-1}(\mathbf{K}_{Z^{(2)}z} - \mathbf{K}_{Z^{(2)}z_0}). \tag{59}$$

This is the estimator of $\mathbb{E}[Y|Z = z] - \mathbb{E}[Y|Z = z_0]$.

**Stage 2.** The optimization problem in **Stage 2** using $\mathcal{D}^{(2)}$ is

$$\min_{H \in \mathcal{L}_2(\mathcal{H}_{X,\boldsymbol{W}}, \Omega_Y)} N_2^{-1}\sum_{i=1}^{N_2}\left\| \hat{G}_2(\psi(z_i^{(2)}) - \psi(z_0)) - H(\hat{G}_1(\psi(z_i^{(2)}) - \psi(z_0)))\right\|^2$$
$$+ \xi\|H\|_{\mathcal{L}_2(\mathcal{H}_{X,\boldsymbol{W}}, \Omega_Y)}^2 + \lambda_3\|H \circ \hat{G}_1\|_{\mathcal{L}_2(\mathcal{H}_Z, \Omega_Y)}^2. \tag{60}$$

Then, the estimation problem reduces to

$$\frac{1}{N_2}\sum_{i=1}^{N_2}(y_i^{(2)} - u_0(z_0) - \boldsymbol{\alpha}^T w(z))^2 + \xi\boldsymbol{\alpha}^T\mathbf{K}_{XX}\boldsymbol{\alpha} + \lambda_3\boldsymbol{\alpha}^T\boldsymbol{\alpha} \tag{61}$$

$$= \frac{1}{N_2}\|\boldsymbol{y}^{(2)} - \boldsymbol{y}^{(2)T}(\mathbf{K}_{Z^{(2)}Z^{(2)}} + N_2\lambda_2\mathbf{I})^{-1}\mathbf{K}_{Z^{(2)}z_0} \tag{62}$$

$$-(\mathbf{K}_{X^{(1)}X^{(1)}}(\mathbf{K}_{Z^{(1)}Z^{(1)}} + N_1\lambda_1\mathbf{I})^{-1}(\mathbf{K}_{Z^{(1)}Z^{(2)}} - \mathbf{K}_{Z^{(1)}z_0}))^T\boldsymbol{\alpha}\|^2 \tag{63}$$

$$+\xi\boldsymbol{\alpha}^T\mathbf{K}_{(X,\boldsymbol{W})^{(1)}(X,\boldsymbol{W})^{(1)}}\boldsymbol{\alpha} + \lambda_3\boldsymbol{\alpha}^T\boldsymbol{\alpha}, \tag{64}$$

and the solution to this optimization problem can be represented as

$$\hat{\boldsymbol{\alpha}} = (\hat{\mathbf{O}}\hat{\mathbf{O}}^T + N_2\xi\mathbf{K}_{(X,\boldsymbol{W})^{(1)}(X,\boldsymbol{W})^{(1)}} + N_2\lambda_3\mathbf{I})^{-1}\hat{\mathbf{O}} \tag{65}$$

$$\times(\boldsymbol{y}^{(2)} - \boldsymbol{y}^{(2)T}(\mathbf{K}_{Z^{(2)}Z^{(2)}} + N_2\lambda_2\mathbf{I})^{-1}\mathbf{K}_{Z^{(2)}z_0}) \tag{66}$$

$$\hat{\mathbf{O}} = \mathbf{K}_{(X,\boldsymbol{W})^{(1)}(X,\boldsymbol{W})^{(1)}}(\mathbf{K}_{Z^{(1)}Z^{(1)}} + N_1\lambda_1\mathbf{I})^{-1}(\mathbf{K}_{Z^{(1)}Z^{(2)}} - \mathbf{K}_{Z^{(1)}z_0}). \tag{67}$$

Finally, RKHS CAPCE estimator of $(x, \boldsymbol{w})$ becomes $\hat{\mathbb{E}}[\partial_x Y_x|\boldsymbol{w}] = \hat{\boldsymbol{\alpha}}^T\mathbf{K}_{(X,\boldsymbol{W})(x,\boldsymbol{w})}$. $\qquad\square$

# B  CONSISTENCY OF SIEVE CAPCE ESTIMATOR

In this section, we show that sieve CAPCE estimator is consistent under assumptions similar to those guaranteeing the consistency of sieve NTSLS [Newey and Powell, 2003].

### NOTATIONS

We introduce the notations for the assumptions.

**Conditional Moment Restrictions.** The estimation problem reduces to the problem called conditional moment restrictions, and properties of the estimator are well studied [Newey and Powell, 2003, Ai and Chen, 2003], and it is widely used in machine learning fields [Kato et al., 2022]. Since $\mathbb{E}[Y|Z = z_0] - \mathbb{E}[Y|Z] = \mathbb{E}[\mathbb{E}[Y|Z = z_0] - Y|Z]$ and $\mathbb{E}[\mathbb{1}_{X \leq x, \boldsymbol{W}=\boldsymbol{w}}|Z = z] - \mathbb{E}[\mathbb{1}_{X \leq x, \boldsymbol{W}=\boldsymbol{w}}|Z = z_0] = \mathbb{E}[\mathbb{1}_{X \leq x, \boldsymbol{W}=\boldsymbol{w}} - \mathbb{E}[\mathbb{1}_{X \leq x, \boldsymbol{W}=\boldsymbol{w}}|Z = z_0]|Z = z]$, Theorem 3.1 reduces to

$$\mathbb{E}\left[(Y_{X_{z_0}} - Y) - \mathfrak{g}(X, X_{z_0}, \boldsymbol{W}, g)\Big|Z = z\right] = 0 \tag{68}$$

where $\mathfrak{g}(X, X_{z_0}, \boldsymbol{W}, g) = \int_{\Omega_{\boldsymbol{W}}}\int_{\Omega_X}\{\mathbb{1}_{X \leq x, \boldsymbol{W}=\boldsymbol{w}} - \mathbb{1}_{X_{z_0} \leq x, \boldsymbol{W}=\boldsymbol{w}}\}g(X, \boldsymbol{W})dxd\boldsymbol{w}$. We denote residual function $\rho(Y, Y_{X_{z_0}}, X, X_{z_0}, \boldsymbol{W}, g) = (Y_{X_{z_0}} - Y) - \mathfrak{g}(X, \boldsymbol{W}, g))$. Then, the integral equation can be represented by $\mathbb{E}[\rho(Y, Y_{X_{z_0}}, X, X_{z_0}, \boldsymbol{W}, g)|Z] = 0$.

**Consistency of Sieve CAPCE Estimator.** First, we show consistency without compactness restriction. The Sieve CAPCE estimator reduces to the general form of the conditional moment restrictions method, which is well-studied in [Newey and Powell, 2003], as below:

$$\hat{g} = \arg\min_{g \in \mathcal{G}}\sum_{i=1}^{N}\frac{1}{N}\hat{\rho}(z_i, g)^2, \tag{69}$$

where $\hat{\rho}(z_i, g) = \hat{c}_i - \hat{\boldsymbol{d}}_i\boldsymbol{\beta}$, and $\hat{\rho}(z_i, g)$ can be considered as the estimators of $\mathbb{E}[\rho(Y, Y_{X_{z_0}}, X, X_{z_0}, \boldsymbol{W}, g)|Z = z_i]$.

### ASSUMPTIONS

We make the following consistency assumptions introduced in [Newey and Powell, 2003]. We denote $\mathcal{G}_S = \{g \in \mathcal{G} : \|\mathfrak{g}_0(x, \boldsymbol{w})\|_{\widetilde{W}^{l,2}}^2 \leq B_S\}$, and $\overline{\mathcal{G}_S}$ is a closure of $\mathcal{G}_S$.

**Assumption B.1** (Uniqueness of g). $g_0 \in \mathcal{G}_S$ is the only $g \in \mathcal{G}_S$ satisfying $\mathbb{E}[\rho(Y, Y_{X_{z_0}}, X, X_{z_0}, \boldsymbol{W}, g)|Z = z] = 0$.

**Assumption B.2** (Completeness of **Stage 1.**). Taking limits $P \to \infty$, $N \to \infty$ with $P/N \to 0$, there exists $\boldsymbol{\pi}_P$ with $\mathbb{E}[\{b(z) - \boldsymbol{q}(z)^T\boldsymbol{\pi}_P\}^2] \to 0$ for any $b(z)$ with $\mathbb{E}[b(z)^2] < \infty$.

The above assumption is for the completeness of parameter space used in Stage 1.

**Assumption B.3** (Boundedness of $\rho$). $\mathbb{E}[\|\rho(Y, Y_{X_{z_0}}, X, X_{z_0}, \boldsymbol{W}, g)\|^2 | Z]$ *is bounded and there exists* $M(Y, Y_{X_{z_0}}, X, X_{z_0}, \boldsymbol{W})$, $\nu > 0$ *such that for all* $\tilde{g}, g \in \overline{\mathcal{G}_S}$, $\|\rho(Y, Y_{X_{z_0}}, X, X_{z_0}, \boldsymbol{W}, \tilde{g}) - \rho(Y, Y_{X_{z_0}}, X, X_{z_0}, \boldsymbol{W}, g)\| \leq M(Y, Y_{X_{z_0}}, X, X_{z_0}, \boldsymbol{W})\|\tilde{g} - g\|_{W^{l,2}}^{\nu}$ *and* $\mathbb{E}[M(Y, Y_{X_{z_0}}, X, X_{z_0}, \boldsymbol{W})^2 | Z]$ *is bounded.*

The above assumption is for the boundness of the parameters used in stage 2.

Let $\mathcal{W}$ denote the domain of $\mathfrak{g}(x, \boldsymbol{w}, g)$.

**Assumption B.4** (Openness and Convexness of Restricted Parameter Space). $\mathcal{W}$ *is open and convex.*

The following lemma is shown in [Newey and Powell, 2003]:

**Lemma B.1.** *If (i) $\Theta$ is a compact subset of a space with norm $\|\theta\|$: (ii) $\hat{Q}(\theta) \to_p Q(\theta)$ for all $\theta \in \Theta$: (iii) there is $v > 0$ and $B_n O_p(1)$ such that for all $\tilde{\theta}, \theta \in \Theta$, $|\hat{Q}(\theta) - \hat{Q}(\tilde{(\theta)})| \leq B_n \Delta^v = B_n \epsilon / 2M \leq \epsilon / 2$ with a positive probability, then $Q(\theta)$ is continuous and $\sup_{\theta \in \Theta} |\hat{Q}(\theta) - Q(\theta)| \to_p 0$.*

**Theorem 4.1.** *Under SCM $\mathcal{M}_{IV}$ and Assumptions 3.1, 3.2, 4.1, 4.2, 4.3, B.1, B.2, B.3, and B.4, letting $P \to \infty$ and $J \to \infty$, then $\|\hat{g} - g_0\|_{W^{l,\infty}} \xrightarrow{p} 0$.*

*Proof.* From the Assumption B.2 and B.4, the parameter space is compact subset. From the Assumption B.3, the following relation is satisfied:

$$|\rho(Y, Y_{X_{z_0}}, X, X_{z_0}, \boldsymbol{W}, \tilde{g}) - \rho(Y, Y_{X_{z_0}}, X, X_{z_0}, \boldsymbol{W}, g)| \tag{70}$$
$$\leq M(Y, Y_{X_{z_0}}, X, X_{z_0}, \boldsymbol{W})\|\tilde{g} - g\|_{W^{l,2}}^{\nu} \tag{71}$$

From the Lemma B.1,

$$\|\tilde{g} - g\|_{W^{l,\infty}} \to_p 0. \tag{72}$$

From Assumption B.1, the limits of $\tilde{g}$ is $g_0$. $\qquad \square$

From the definition of $W^{l,\infty}$, this theorem means uniform convergence.

# C   RATE OF CONVERGENCE OF SIEVE CAPCE ESTIMATOR

**NOTATIONS**

In this section, we explain the notations used in the assumptions for Theorem 4.2 and Theorem 4.4. Denote the estimation problem

$$\inf_{g \in \mathcal{G}} \mathbb{E}\left[\mathfrak{g}(X, X_{z_0}, \boldsymbol{W}, g)^2\right] \tag{73}$$

and introduce norm $\|\cdot\|_A$ as below:

$$\|g_1 - g_0\|_A = \sqrt{\mathbb{E}\left[\left(\frac{d\mathfrak{g}(X, X_{z_0}, \boldsymbol{W}, g_0)}{dg}\right)^2\right]} \tag{74}$$

where

$$\frac{d\rho(Z, g_0)}{dg}[g - g_0] = \frac{d\rho(Z, (1-\tau)g_0 + \tau g)}{d\tau} \text{ a.s. } Z \tag{75}$$

$$\frac{d\rho(Z, g_0)}{dg}[g_1 - g_2] = \frac{d\rho(Z, g_0)}{dg}[g_1 - g_0] - \frac{d\rho(Z, g_0)}{dg}[g_2 - g_0] \tag{76}$$

$$\frac{d\mathfrak{g}(X, X_{z_0}, \boldsymbol{W}, g_0)}{dg} = \mathbb{E}\left[\frac{d\rho(Z, g_0)}{dg}[g_1 - g_2]\Big|\{X, X_{z_0}, \boldsymbol{W}\}\right]. \tag{77}$$

These derivatives are called "pathwise derivatives." See [Ai and Chen, 2003] for details.

To evaluate the rate of convergence, we denote the number of the basis functions depending on sample size be $J_N$ and $P_N$. Note that $N \to \infty$ implies $J_N \to \infty$ and $P_N \to \infty$. We use more basis functions, $\boldsymbol{q}^{P_N} = (q^1, \ldots, q^{P_N})$, as the sample size grows for the stage 1.

**ASSUMPTIONS**

We make the following assumptions.

**Assumption C.1** (Compactness of Domain). *$\Omega_{(X, X_{z_0}, \boldsymbol{W})}$ is compact with non empty interior.*

**Assumption C.2** (Order of Convergence of Stage 1). *For any $h \in \mathcal{G}_S$ with $\kappa > (1 + d)/2$, there exists $\boldsymbol{q}^{P_N}(X, X_{z_0}, \boldsymbol{W})^T \boldsymbol{\pi}_{P_N} \in \mathcal{G}_S$, where $\boldsymbol{\pi}_{P_N}$ is $P_N$ vector, such that $\sup_{(X, X_{z_0}, \boldsymbol{W}) \in \Omega_{(X, X_{z_0}, \boldsymbol{W})}} |h(X, X_{z_0}, \boldsymbol{W}) - \boldsymbol{q}^{P_N}(X, X_{z_0}, \boldsymbol{W})^T \boldsymbol{\pi}_{P_N}| = \mathcal{O}(P_N^{-\kappa/(1+d)})$ and $P_N^{-\kappa/(1+d)} = o(N^{-1/4})$.*

The above assumption guarantees the order of convergence of regression (basis functions) used in Stage 1.

**Assumption C.3** (Order of Convergence of Stage 2). *There is a constant $\mu_1 > 0$ such that for any $g \in \mathcal{G}$, there is $\Pi g \in \mathcal{G}$ satisfying $\|\Pi g - g\| = \mathcal{O}(J_N^{-\kappa/(1+d)})$ and $J_N^{-\kappa/(1+d)} = o(N^{-1/4})$. $\Pi$ is the projections to $\mathcal{G}$.*

The above assumption guarantees the order of convergence of regression (basis functions) used in Stage 2.

**Assumption C.4** (Envelope condition). *Each element of $\rho(Z, g)$ satisfies the envelope condition in $g \in \mathcal{G}$; and, each element of $\rho(Z, g) \in \mathcal{G}_S$ with $\kappa > (1 + d)/2$.*

The envelope condition is shown in [Milgrom and Segal, 2002].

Denote $\xi_N = \sup_{(X, X_{z_0}, \boldsymbol{W})} \|\boldsymbol{q}^{P_N}(X, X_{z_0}, \boldsymbol{W})\|$.

**Assumption C.5** (Condition of $J_N$). *$J_N \times ln(N) \times \xi_N \times N^{-1/2} = o(1)$*

We denote $N(\epsilon^{1/k}, \mathcal{G}, \|\cdot\|_{W^{l,2}})$ as the minimal number of radius $\delta$ covering ball of $\mathcal{G}$.

**Assumption C.6** (Condition of $J_N$). *$ln[N(\epsilon^{1/k}, \mathcal{G}, \|\cdot\|_{W^{l,2}})] \leq const. \times J_N \times ln(J_N/\epsilon)$*

These assumptions show how to make the models complex depending on sample size.

**Assumption C.7** (Convexness of Parameter Space). *$\mathcal{G}$ is convex in $g$, and $\rho(Z, g)$ is pathwise differentiable at $g$; and, for some $c_1, c_2 > 0$,*

$$c_1 \mathbb{E}[\hat{\rho}(Z, g)^2] \leq \|\hat{g} - g\|^2 \leq c_2 \mathbb{E}[\hat{\rho}(Z, g)^2] \tag{78}$$

*holds for all $\hat{g} \in \mathcal{G}$ with $\|\hat{g} - g\|_{W^{l,2}}^2 = o(1)$*

The following lemma holds [Ai and Chen, 2003]:

**Lemma C.1.** *Under Assumptions C.1, C.2, C.3, C.4, C.5, C.6, and C.7, (i) $\hat{L}_N(g) - L_N(g) = o_p(N^{-1/4})$ uniformly over $g \in \mathcal{G}$; and (ii) $\hat{L}_N(g) - \hat{L}_N(g_0) - \{L_N(g) - L_N(g_0)\} = o_p(\tau_N N^{-1/4})$ uniformly over $g \in \mathcal{G}$ with $\|g - g_0\| \leq o(\tau_N)$, where $\tau_N = N^{-\tau}$ with $\tau \leq 1/4$.*

**Theorem 4.2.** *Under SCM $\mathcal{M}_{IV}$ and Assumptions 3.1, 3.2, 4.1, 4.2, 4.3, C.1, C.2, C.3, C.4, C.5, C.6, and C.7, setting $N = N_1 = N_2$, then $\|\hat{g} - g_0\|_A = o_p(N^{-1/4})$.*

*Proof.* Let

$$\hat{L}_N(g) = -\frac{1}{2N}\hat{\mathfrak{g}}(X, X_{z_0}, \boldsymbol{W}.g)^2, \quad L_N(g) = -\frac{1}{2N}\mathfrak{g}(X, X_{z_0}, \boldsymbol{W}, g)^2. \tag{79}$$

Then, Lemma C.1 implies

$$\hat{L}_N(g) - \hat{L}_N(g_0) - \{L_N(g) - L_N(g_0)\} = o_p(N^{-1/4}) \tag{80}$$

and this proves

$$\|\hat{g} - g_0\| = o_p(N^{-1/4}). \tag{81}$$

$\square$

# D   CONSISTENCY OF PARAMETRIC CAPCE ESTIMATOR

In this section, we show the consistency property of parametric CAPCE estimator. We denote the functional space $\mathcal{G}$ be $\{g \in \mathcal{G} : g(x, \boldsymbol{w}) = \sum_{k=1}^{K} \gamma_k \theta_k(x, \boldsymbol{w})\}$.

**Consistency of Parametric CAPCE Estimator.** First, we show consistency without compactness restriction. The Parametric CAPCE estimator reduces to the general form of the conditional moment restrictions method, which is well-studied in [Newey and Powell, 2003], as below:

$$\hat{\boldsymbol{\gamma}} = \arg\min_{\boldsymbol{\gamma}} \sum_{i=1}^{N} \frac{1}{N}\hat{\rho}(z_i, \boldsymbol{\gamma})^2, \tag{82}$$

where $\hat{\rho}(z_i, \boldsymbol{\gamma}) = \hat{c}_i - \hat{\boldsymbol{e}}_i\boldsymbol{\gamma}$. $\hat{\rho}(z_i, \boldsymbol{\gamma})$ can be considered as the estimators of $\mathbb{E}[\rho(Y, Y_{X_{z_0}}, X, X_{z_0}, \boldsymbol{W}, \boldsymbol{\gamma})|Z = z_i]$.

## ASSUMPTIONS

We make the following assumptions introduced in [Newey and Powell, 2003]. We denote $\mathcal{G}_P = \{\boldsymbol{\gamma}^T\boldsymbol{\gamma} \leq B_P\}$, and $\overline{\mathcal{G}_P}$ is the closure of $\mathcal{G}_P$.

**Assumption D.1** (Uniqueness of $g$). *$\boldsymbol{\gamma} \in \mathcal{G}_P$ is the only $\boldsymbol{\gamma} \in \mathcal{G}_P$ satisfying $\mathbb{E}[\rho(Y, Y_{X_{z_0}}, X, X_{z_0}, \boldsymbol{W}, \boldsymbol{\gamma})|Z = z] = \boldsymbol{0}$.*

**Assumption D.2** (Completeness of $\boldsymbol{q}$). *Taking limits $P \to \infty$, $N \to \infty$ with $P/N \to 0$, there exists $\boldsymbol{\pi}_P$ with $\mathbb{E}[\{b(z) - \boldsymbol{q}(z)^T\boldsymbol{\pi}_P\}^2] \to 0$ for any $b(z)$ with $\mathbb{E}[b(z)^2] < \infty$.*

**Assumption D.3** (Boundedness of $\rho$). *$\mathbb{E}[\|\rho(Y, Y_{X_{z_0}}, X, X_{z_0}, \boldsymbol{W}, \boldsymbol{\gamma})\|^2|Z]$ is bounded and there exists $M(Y, Y_{X_{z_0}}, X, X_{z_0}, \boldsymbol{W})$, $\nu > 0$ such that for all $\tilde{\boldsymbol{\gamma}}, \boldsymbol{\gamma} \in \overline{\mathcal{G}_P}$, $\|\rho(Y, Y_{X_{z_0}}, X, X_{z_0}, \boldsymbol{W}, \tilde{\boldsymbol{\gamma}}) - \rho(Y, Y_{X_{z_0}}, X, X_{z_0}, \boldsymbol{W}, g)\| \leq M(Y, Y_{X_{z_0}}, X, X_{z_0}, \boldsymbol{W})\|\tilde{\boldsymbol{\gamma}} - \boldsymbol{\gamma}\|^\nu$ and $\mathbb{E}[M(Y, Y_{X_{z_0}}, X, X_{z_0}, \boldsymbol{W})^2|Z]$ is bounded.*

Let $\mathcal{W}$ denote the domain of $\mathfrak{g}(x, \boldsymbol{w}, \boldsymbol{\gamma})$.

**Assumption D.4** (Openness and Convexness of Restricted Parameter Space). *$\mathcal{W}$ is open and convex.*

**Theorem 4.3.** *Under SCM $\mathcal{M}_{IV}$ and Assumptions 3.1, 3.2, 4.4, D.1, D.2, D.3, and D.4, letting $P \to \infty$, then $\|\hat{\boldsymbol{\gamma}} - \boldsymbol{\gamma}\| \xrightarrow{p} 0$.*

*Proof.* From the Assumption D.2 and D.4, the parameter space is compact subset. From the Assumption D.3, the following relation is satisfied:

$$|\rho(Y, Y_{X_{z_0}}, X, X_{z_0}, \boldsymbol{W}, \tilde{\boldsymbol{\gamma}}) - \rho(Y, Y_{X_{z_0}}, X, X_{z_0}, \boldsymbol{W}, \boldsymbol{\gamma})| \tag{83}$$
$$\leq M(Y, Y_{X_{z_0}}, X, X_{z_0}, \boldsymbol{W})\|\tilde{\boldsymbol{\gamma}} - \boldsymbol{\gamma}\|^\nu \tag{84}$$

From Lemma B.1,

$$\|\tilde{\boldsymbol{\gamma}} - \boldsymbol{\gamma}\| \to_p 0. \tag{85}$$

From Assumption D.1, the limits is $\boldsymbol{\gamma}_0$.

$\square$

# E  RATE OF CONVERGENCE OF PARAMETRIC CAPCE ESTIMATOR

## ASSUMPTIONS

We make the following assumptions.

**Assumption E.1** (Compactness of Domain). $\Omega_{(X,X_{z_0},\boldsymbol{W})}$ *is compact with non empty interior.*

**Assumption E.2** (Order of Convergence of Stage 1). *For any* $h \in \mathcal{G}_P$ *with* $\kappa > (1+d)/2$, *there exists* $\boldsymbol{q}^{P_N}(X, X_{z_0}, \boldsymbol{W})^T \boldsymbol{\pi}_{P_N} \in \mathcal{G}_P$, *where* $\boldsymbol{\pi}_{P_N}$ *is* $P_N$ *vector, such that* $\sup_{(X,X_{z_0},\boldsymbol{W})\in\Omega_{(X,X_{z_0},\boldsymbol{w})}} |h(X, X_{z_0}, \boldsymbol{W}) - \boldsymbol{q}^{P_N}(X, X_{z_0}, \boldsymbol{W})^T \boldsymbol{\pi}_{P_N}| = \mathcal{O}(P_N^{-\kappa/(1+d)})$ *and* $P_N^{-\kappa/(1+d)} = o(N^{-1/4})$.

**Assumption E.3** (Order of Convergence of Stage 2). *There is a constant* $\mu_1 > 0$ *such that for any* $\boldsymbol{\gamma} \in \mathcal{G}_P$, *there is* $\Pi\boldsymbol{\gamma} \in \mathcal{G}_P$ *satisfying* $\|\Pi\boldsymbol{\gamma} - \boldsymbol{\gamma}\| = \mathcal{O}(1)$.

**Assumption E.4** (Envelope condition). *Each element of* $\rho(Z, \boldsymbol{\gamma})$ *satisfies the envelope condition in* $\boldsymbol{\gamma} \in \mathcal{G}_P$; *and, each element of* $\rho(Z, \boldsymbol{\gamma}) \in \mathcal{G}_P$ *with* $\kappa > (1+d)/2$, *for all* $\boldsymbol{\gamma} \in \mathcal{G}_P$.

The envelope condition is shown in [Milgrom and Segal, 2002].

**Assumption E.5** (Convexness of Parameter Space). $\mathcal{G}_P$ *is convex in* $\boldsymbol{\gamma}$, *and* $\rho(Z, \boldsymbol{\gamma})$ *is pathwise differentiable at* $\boldsymbol{\gamma}$; *and, for some* $c_1, c_2 > 0$,

$$c_1 \mathbb{E}[\hat{\rho}(Z, \boldsymbol{\gamma})^2] \leq \|\hat{\boldsymbol{\gamma}} - \boldsymbol{\gamma}\|^2 \leq c_2 \mathbb{E}[\hat{\rho}(Z, \boldsymbol{\gamma})^2] \tag{86}$$

*holds for all* $\hat{\boldsymbol{\gamma}} \in \mathcal{G}_P$ *with* $\|\hat{\boldsymbol{\gamma}} - \boldsymbol{\gamma}\|^2 = o(1)$

The following lemma holds [Ai and Chen, 2003]:

**Lemma E.1.** *Under Assumptions E.1, E.2, E.3, E.4, and E.5, (i)* $\hat{L}_N(\boldsymbol{\gamma}) - L_N(\boldsymbol{\gamma}) = o_p(N^{-1/4})$ *uniformly over* $\boldsymbol{\gamma}$; *and (ii)* $\hat{L}_N(\boldsymbol{\gamma}) - \hat{L}_N(\boldsymbol{\gamma}_0) - \{L_N(\boldsymbol{\gamma}) - L_N(\boldsymbol{\gamma}_0)\} = o_p(\tau_N N^{-1/4})$ *uniformly over* $\boldsymbol{\gamma}$ *with* $\|\boldsymbol{\gamma} - \boldsymbol{\gamma}_0\| \leq o(\tau_N)$, *where* $\tau_N = N^{-\tau}$ *with* $\tau \leq 1/4$.

**Theorem 4.4.** *Under SCM* $\mathcal{M}_{IV}$ *and Assumptions 3.1, 3.2, 4.4, E.1, E.2, E.3, E.4, and E.5, setting* $N = N_1 = N_2$, *then* $\|\hat{\boldsymbol{\gamma}} - \boldsymbol{\gamma}\| = o_p(N^{-1/4})$.

*Proof.* Let

$$\hat{L}_N(\boldsymbol{\gamma}) = -\frac{1}{2N}\hat{\mathfrak{g}}(X, X_{z_0}, \boldsymbol{W}, \boldsymbol{\gamma})^2, \quad L_N(g) = -\frac{1}{2N}\mathfrak{g}(X, X_{z_0}, \boldsymbol{W}, \boldsymbol{\gamma})^2. \tag{87}$$

Then, Lemma E.1 implies

$$\hat{L}_N(\boldsymbol{\gamma}) - \hat{L}_N(\boldsymbol{\gamma}_0) - \{L_N(\boldsymbol{\gamma}) - L_N(\boldsymbol{\gamma}_0)\} = o_p(N^{-1/4}) \tag{88}$$

and this proves

$$\|\hat{\boldsymbol{\gamma}} - \boldsymbol{\gamma}_0\| = o_p(N^{-1/4}). \tag{89}$$

$\square$

# F  PROPERTIES OF RKHS CAPCE ESTIMATOR

We show the consistency and rate of convergence of RKHS CAPCE estimator following [Singh et al., 2019] when $\lambda_3$ is 0.

## NOTATIONS

We use the integral operator notations from the kernel methods literature. $\mathcal{L}_2(\Omega_Z, \mathfrak{p}_Z)$ denotes a $\mathcal{L}_2$ integrable function from $\Omega_Z$ to $\Omega_Y$ with respect to measure $\mathfrak{p}_Z$.

**Definition 2.** *The stage 1 operators are*

$$S_1^* : \mathcal{H}_Z \to \mathcal{L}_2(\Omega_Z, \mathfrak{p}_Z), l \mapsto \langle l, \psi(\cdot) \rangle_{\mathcal{H}_Z} \tag{90}$$

$$S_1 : \mathcal{L}_2(\Omega_Z, \mathfrak{p}_Z) \to \mathcal{H}_Z, \tilde{l} \mapsto \int \psi(z) \tilde{l}(z) \mathfrak{p}_Z(z) dz \tag{91}$$

*and $T_1 = S_1^* \circ S_1$ is the uncentered covariance operator. The details of the theory of vector-valued RKHS are shown in [Singh et al., 2019].*

In addition, we denote

**Definition 3.**

$$G_{1\rho} = \arg\min \mathcal{E}_1(E), \mathcal{E}_1 = \mathbb{E}[\pi(X, \boldsymbol{W}) - G_1(\psi(Z))]^2_{\mathcal{H}_{X,\boldsymbol{W}}}, \tag{92}$$

$$G_{1\lambda} = \arg\min \mathcal{E}_1(G_1), \mathcal{E}_1 = \mathbb{E}[\pi(X, \boldsymbol{W}) - G_1(\psi(Z))]^2_{\mathcal{H}_{X,\boldsymbol{W}}} + \lambda \|G_1\|^2_{\mathcal{L}_2(\mathcal{H}_Z, \mathcal{H}_{X,\boldsymbol{W}})}, \tag{93}$$

$$\hat{G}_{1\lambda} = \arg\min \mathcal{E}_1(G_1), \mathcal{E}_1 = \hat{\mathbb{E}}[\pi(X, \boldsymbol{W}) - G_1(\psi(Z))]^2_{\mathcal{H}_{X,\boldsymbol{W}}} + \lambda \|G_1\|^2_{\mathcal{L}_2(\mathcal{H}_Z, \mathcal{H}_{X,\boldsymbol{W}})}, \tag{94}$$

$$G_{2\rho} = \arg\min \mathcal{E}_1(E), \mathcal{E}_1 = \mathbb{E}[Y - G_2(\psi(Z))]^2, \tag{95}$$

$$G_{2\lambda} = \arg\min \mathcal{E}_1(G_2), \mathcal{E}_1 = \mathbb{E}[Y - G_2(\psi(Z))]^2 + \lambda \|G_2\|^2_{\mathcal{L}_2(\mathcal{H}_Z, \Omega_Y)}, \tag{96}$$

$$\hat{G}_{2\lambda} = \arg\min \mathcal{E}_1(G_2), \mathcal{E}_1 = \hat{\mathbb{E}}[Y - G_2(\psi(Z))]^2 + \lambda \|G_2\|^2_{\mathcal{L}_2(\mathcal{H}_Z, \Omega_Y)}. \tag{97}$$

**Definition 4.** *The stage 2 operators are*

$$S_2^* : \mathcal{L}_2(\mathcal{H}_Z, \mathcal{H}_{X,\boldsymbol{W}}) \to \mathcal{L}_2(\mathcal{H}_{X,\boldsymbol{W}}, \mathfrak{p}_{\mathcal{H}_{X,\boldsymbol{W}}}), H \mapsto \Omega^*_{(\cdot)} H \tag{98}$$

$$S_2 : \mathcal{L}_2(\mathcal{H}_{X,\boldsymbol{W}}, \mathfrak{p}_{\mathcal{H}_{X,\boldsymbol{W}}}) \to \mathcal{L}_2(\mathcal{H}_Z, \mathcal{H}_{X,\boldsymbol{W}}), \tag{99}$$

$$\tilde{H} \mapsto \int \Omega_{\mu(z) - \mu(z_0)} \circ \tilde{H} \{\mu(z) - \mu(z_0)\} \mathfrak{p}_{\mathcal{H}_{X,\boldsymbol{W}}}(\mu(z)) \tag{100}$$

*and $T_2 = S_2^* \circ S_2$ is the uncentered covariance operator.*

**Definition 5.** *We denote*

$$H_\rho = \arg\min \mathcal{E}(H), \mathcal{E}(H) = \mathbb{E}[Y - \mu_2(z_0) - H(\mu(Z) - \mu(z_0))]^2_{\mathcal{H}_{X,\boldsymbol{W}}}, \tag{101}$$

$$H_\xi = \arg\min \mathcal{E}_\xi(H), \tag{102}$$

$$\mathcal{E}(H) = \mathbb{E}[Y - \mu_2(z_0) - H(\mu(Z) - \mu(z_0))]^2_{\mathcal{H}_{X,\boldsymbol{W}}} + \xi \|H\|^2_{\mathcal{L}_2(\mathcal{H}_{X,\boldsymbol{W}}, \Omega_Y)}, \tag{103}$$

$$\hat{H}_\xi = \arg\min \hat{\mathcal{E}}_\xi(H), \tag{104}$$

$$\hat{\mathcal{E}}(H) = \hat{\mathbb{E}}[Y - \mu_2(z_0) - H(\mu(Z) - \mu(z_0))]^2_{\mathcal{H}_{X,\boldsymbol{W}}} + \xi \|H\|^2_{\mathcal{L}_2(\mathcal{H}_{X,\boldsymbol{W}}, \Omega_Y)}. \tag{105}$$

## ASSUMPTIONS

Next, we show assumptions for Theorem 4.5.

**Assumption F.1** (Restriction for the domains). *Suppose that $\Omega_{X,\boldsymbol{W}}$ and $\Omega_Z$ are Polish spaces, i.e., separable and completely metrizable topological spaces.*

**Assumption F.2** (Restriction for the feature functions). *Suppose that*

1. *$k_{X,\boldsymbol{W}}$ and $k_{\boldsymbol{Z}}$ are continuous and bounded: $\sup_{x\in\Omega_{X,\boldsymbol{W}}}\|\pi(x,\boldsymbol{w})\|_{\mathcal{H}_{X,\boldsymbol{W}}} \leq Q$ and $\sup_{z\in\Omega_Z}\|\psi(z)\|_{\mathcal{H}_{\boldsymbol{Z}}} \leq \kappa$.*
2. *$\pi$ and $\psi$ are measurable.*
3. *$k_{X,\boldsymbol{W}}$ is characteristic.*

**Assumption F.3** (Uniqueness). *Suppose that $G_{1\rho} \in \mathcal{L}_2(\mathcal{H}_Z, \mathcal{H}_{\boldsymbol{Z}})$, then $\mathcal{E}_1(G_{1\rho}) = \inf_{G_1\in\mathcal{H}_Z}\mathcal{E}_1(G_1)$. Furthermore, suppose that $G_{2\rho} \in \mathcal{L}_2(\mathcal{H}_Z, \mathcal{H}_{\boldsymbol{Z}})$, then $\mathcal{E}_1(G_{2\rho}) = \inf_{G_2\in\mathcal{H}_Z}\mathcal{E}_1(G_2)$.*

**Assumption F.4** (Boundness of stage 1). *Fix $\zeta_1, \zeta_2 \leq \infty$. For given $c_1, c_2 \in (1, 2]$, define the prior $\mathcal{P}(\zeta_1, c_1)$ and $\mathcal{P}(\zeta_2, c_2)$ as the set of the probability distributions on $\Omega_{X,\boldsymbol{W}} \times \Omega_Z$ such that a range space assumption is satisfied: $\exists C_1 \in \mathcal{L}_2(\mathcal{H}_Z, \mathcal{H}_{X,\boldsymbol{W}})$ such that $G_{1\rho} = T_1^{\frac{c_1-1}{2}} \circ C_1$ and $\|C_1\|^2_{\mathcal{L}_2(\mathcal{H}_Z, \mathcal{H}_{X,\boldsymbol{W}})} \leq \zeta_1$, and $\exists C_2 \in \mathcal{L}_2(\mathcal{H}_Z, \Omega_Y)$ such that $G_{2\rho} = T_1^{\frac{c_2-1}{2}} \circ C_2$ and $\|C_2\|^2_{\mathcal{L}_2(\mathcal{H}_Z, \Omega_Y)} \leq \zeta_2$.*

**Lemma F.1** (Rate of convergence of stage 1 (A)). *Make Assumptions F.1, F.2, F.3 and F.4. For all $\delta \in (0, 1)$, the following holds w.p. $1 - \delta$:*

$$\|\hat{G}_{1\lambda} - G_{1\rho}\|_{\mathcal{L}_2(\mathcal{H}_Z, \mathcal{H}_{X,\boldsymbol{W}})}$$
$$\leq \frac{\sqrt{\zeta_1}(c_1+1)}{4^{\frac{1}{c_1+1}}}\left(\frac{4\kappa(Q + \kappa\|G_{1\rho}\|_{\mathcal{L}_2(\mathcal{H}_Z, \mathcal{H}_{X,\boldsymbol{W}})})ln(2/\delta)}{\sqrt{n\zeta_1}(c_1-1)}\right) \tag{106}$$

**Lemma F.2** (Rate of convergence of stage 1 (B)). *Make Assumptions F.1, F.2, F.3 and F.4. For all $\delta \in (0, 1)$, the following holds w.p. $1 - \delta$:*

$$\|\hat{G}_{2\lambda} - G_{2\rho}\|_{\mathcal{L}_2(\mathcal{H}_Z, \Omega_Z)}$$
$$\leq \frac{\sqrt{\zeta_2}(c_2+1)}{4^{\frac{1}{c_2+1}}}\left(\frac{4\kappa(Q + \kappa\|G_{2\rho}\|_{\mathcal{L}_2(\mathcal{H}_Z, \Omega_Z)})ln(2/\delta)}{\sqrt{n\zeta_2}(c_2-1)}\right) \tag{107}$$

The proof is shown in [Singh et al., 2019]. The above lemma implies consistency of Stage 1 (A).

**Assumption F.5** (Restriction of domain). *Suppose that $\Omega_Y$ is a Polish space, i.e., separable and completely metrizable topological spaces.*

**Assumption F.6** (Boundness of stage 2). *Suppose that*

1. *The $\{\Psi_{\mu(z)-\mu(z_0)}\}$ operator family is uniformly bounded in Hilbert-Schmidt norm: $\exists B$ such that $\forall\mu(z)$, $\|\Psi_{\mu(z)-\mu(z_0)}\|^2_{\mathcal{L}_2(\Omega_Z, \mathcal{L}_2(\mathcal{H}_Z, \mathcal{H}_{X,\boldsymbol{W}}))} = Tr(\Psi^*_{\mu(z)-\mu(z_0)} \circ \Psi_{\mu(z)-\mu(z_0)}) \leq B$.*
2. *The $\{\Psi_{\mu(z)-\mu(z_0)}\}$ operator family is Hölder continuous in operator norm: $\exists L > 0$, $\iota \in (0, 1]$ such that $\forall\mu(z), \mu(z')$, $\|\Psi_{\mu(z)-\mu(z_0)} - \Psi_{\mu(z')-\mu(z_0)}\|_{L(\Omega_Z, \mathcal{L}_2(\mathcal{H}_Z, \mathcal{H}_{X,\boldsymbol{W}}))} \leq L\|\mu(z) - \mu(z')\|^\iota_{\mathcal{H}_{X,\boldsymbol{W}}}$.*

**Assumption F.7** (Boundness of stage 2). *Suppose that*

1. *$\langle H_\rho, \cdot \rangle \in \mathcal{L}_2(\mathcal{H}_{X,\boldsymbol{W}}, \Omega_Y)$. Then, $\mathcal{E}(H_\rho) = \inf_{H\in\mathcal{H}_{X,\boldsymbol{W}}}\mathcal{E}(H)$.*
2. *$Y$ is bounded, i.e. $\exists C < \infty$ such that $\|Y\| \leq C$ almost surely.*

**Assumption F.8** (Boundness of stage 2). *Fix $\zeta < \infty$. For given $b \in (1, \infty]$ and $c \in (1, 2]$, define the prior $\mathcal{P}(\zeta, b, c)$ as the set of probability distributions $\mathfrak{p}$ on $\mathcal{H}_{X,\boldsymbol{W}} \times \Omega_Y$ such that*

1. *A range space assumption is satisfied: $\exists C \in \mathcal{L}_2(\mathcal{H}_{X,\boldsymbol{W}}, \Omega_Y)$ such that $H_\rho = T_2^{\frac{c-1}{2}} \circ C$ and $\|C\|^2_{\mathcal{L}_2(\mathcal{H}_{X,\boldsymbol{W}}, \Omega_Y)} \leq \zeta$.*

2. *In the spectral decomposition $T = \sum_{k=1}^{\infty} \lambda_k e_k \langle \cdot, e_k \rangle_{\mathcal{H}_{X,W}}$, where $\{e_k\}_{k=1}^{\infty}$ is a basis of $Ker(T)^{\perp}$, the eigenvalues satisfies $\alpha \leq k^b \lambda_k \leq \beta$ for some $\alpha, \beta > 0$.*

These assumptions are for the boundness of **Stage 2**.

**Lemma F.3.** *Make Assumptions F.1, F.2, F.3, F.4, F.5, F.6, F.7 and F.8. Let $\lambda = N_1^{-\frac{1}{c_1+1}}$, $N_1 = N_2^{\frac{a(c_1+1)}{\iota(c_1-1)}}$, $a > 0$, and $\lambda_3 = 0$. We have*

1. *if $a \leq \frac{b(c+1)}{bc+1}$ then $\mathcal{E}(\hat{H}_{\xi}) - \mathcal{E}(H_{\rho}) = \mathcal{O}_p(N_2^{-\frac{ac}{c+1}})$ with $\xi = N_2^{-\frac{a}{c+1}}$.*

2. *if $a \geq \frac{b(c+1)}{bc+1}$ then $\mathcal{E}(\hat{H}_{\xi}) - \mathcal{E}(H_{\rho}) = \mathcal{O}_p(N_2^{-\frac{bc}{bc+1}})$ with $\xi = N_2^{-\frac{b}{bc+1}}$.*

Lemma F.3 can be proved from the proof of Theorem 4 in [Singh et al., 2019] by subsituting $\mu(z)$ with $\mu(z) - \mu(z_0)$.

**Theorem 4.5.** *Under SCM $\mathcal{M}_{IV}$ and Assumptions 3.1, 3.2, F.1, F.2, F.3, F.4, F.5, F.6, F.7 and F.8, the RKHS CAPCE estimator in (25) converges pointwise to CAPCE when $\lambda_3 = 0$.*

*Proof.* Lemma F.3 implies consistency of RKHS CAPCE estimator by taking limit $N_2 \to \infty$. $\square$

# G  ADDITIONAL INFORMATION ON EXPERIMENTS AND THE APPLICATION

In this section, we give detailed information about the settings of the experiments and additional experimental results.

We note that the choice of the reference point $z_0$ does not affect the consistency results or rate of convergence, but it may affect the variance of the estimator. In our experiments, we take the minimum value of $Z$ as a standard reference point $z_0$. The choice of the reference point $z_0$ did not affect the standard deviation of the estimators much in our experiments.

## G.1  DETAILED SETTINGS OF EXPERIMENTS

We present detailed settings of numerical experiments in the following.

**Setting of P-CAPCE and PTSLS.** We learn the conditional expectations of basis functions $\mathbb{E}[Y|Z = z]$, $\mathbb{E}[X|Z = z]$, $\mathbb{E}[WX|Z = z]$ and $\mathbb{E}[X^2|Z = z]$ by the nonlinear model, $b_0 + b_1 Z + b_2 Z^2$. We used the basis terms $\{1, W, X\}$ for P-CAPCE and $\{1, W, X, WX, X^2\}$ for PTSLS, which match setting (A), and let $z_0 = -1$. Regularize value is determined by test error from $\{1, 10^{-1}, 10^{-2}, 10^{-3}\}$.

**Setting of NTSLS and sieve CAPCE.** We learn the conditional expectations by the nonlinear model, $b_0 + b_1 Z + b_2 Z^2 + b_3 Z^3$, We consider the basis terms $h_p(X) h_q(W)$ for $p = 0, 1, 2$ and $q = 0, 1, 2$, where $h_p$ is Hermite polynomial functions $(h_0(t) = 1, h_1(t) = t, h_2(t) = t^2 - 1$ and $h_3(t) = t^3 - 3t)$, and let $z_0 = -1$. Let $\kappa = 2$ and $l = 1$, and we calculate $\hat{\Lambda}$ by Monte Carlo integration using uniform distribution $(x, w) = (U(-4, 4), U(-2, 2))$, where $\Omega_X \subseteq [-4, 4]$ and $\Omega_X \subseteq [-2, 2]$. Regularize value is determined by test error from $\{1, 10^{-1}, 10^{-2}, 10^{-3}\}$. We estimate CAPCE via differentiating estimated $\mathbb{E}[Y_x|W = w]$.

**Setting of kernel IV and RKHS CAPCE estimator.** We use polynomial kernel function $k_Z(z, z') = (z^T z' + C_1)^{C_2}$ and $k_{X,W}((x, w)(x, w)^T + C_3)^{C_4}$. We select the kernel parameters $(C_1, C_2)$ and $(C_3, C_4)$ from $\{1, 2, 3, 4, 5\} \times \{1, 2, 3, 4, 5\}$, respectively. We select the regularize values $\lambda_1$ and $\lambda_2$ from $\{1, 10^{-1}, 10^{-2}, 10^{-3}\}$, respectively, and $(\lambda_3, \xi)$ is from Cartesian product set $\{100, 10, 1\} \times \{100, 10, 1\}$.

## G.2  ADDITIONAL INFORMATION ON EXPERIMENTAL RESULTS IN THE BODY OF PAPER

**Results: Parametric setting (A).** The basic statistics of estimated coefficients by 100 time simulations of PTSLS and P-CAPCE are shown in Tables 3 and 4. These tables supplement Table 1 in the paper. The true and estimated CAPCE surfaces over $(X, W)$ are shown in Fig. 4.

**Results: Nonparametric setting (B).** The true and estimated CAPCE surfaces over $(X, W)$ are shown in Fig. 5.

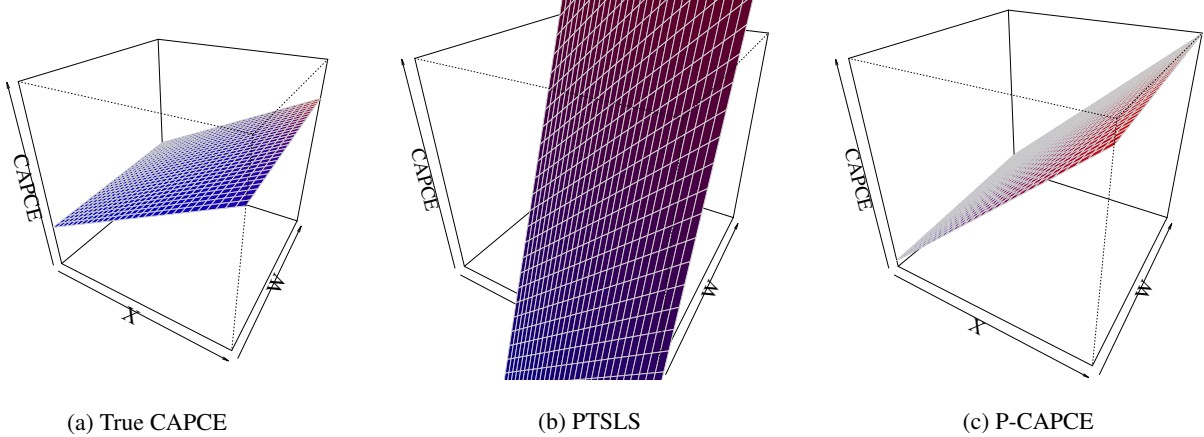

| (a) True CAPCE | (b) PTSLS | (c) P-CAPCE |

Figure 4: Parametric estimated surfaces in setting (A) (Mean, $N = 10000$). X-axis is the value of treatment variable ($X = x$), Y-axis is the value of covariate ($W = w$), and Z-axis is the value of CAPCE.

Table 3: Basic statistics of the P-CAPCE estimator over 1000 runs when $N = 1000$ and $N = 10000$ in setting (A).

| $N = 1000$ | 1 | $W$ | $X$ |
|---|---|---|---|
| True Coeff. | 1 | 1 | 20 |
| Min. | -51.515 | -18.715 | -116.480 |
| 1st Qu. | -10.044 | 0.160 | -10.809 |
| Median | -1.849 | 3.035 | 17.691 |
| 3rd Qu. | 7.926 | 12.239 | 50.007 |
| Max. | 40.458 | 109.791 | 213.306 |
| Mean | -1.651 | 10.383 | 19.293 |
| SD | 14.707 | 22.309 | 50.957 |

| $N = 10000$ | 1 | $W$ | $X$ |
|---|---|---|---|
| True Coeff. | 1 | 1 | 20 |
| Min. | -13.466 | -2.302 | -23.863 |
| 1st Qu. | -2.324 | -0.157 | 10.908 |
| Median | 1.138 | 0.543 | 22.075 |
| 3rd Qu. | 4.643 | 1.568 | 29.509 |
| Max. | 14.988 | 11.017 | 59.559 |
| Mean | 1.226 | 0.963 | 19.971 |
| SD | 5.380 | 2.124 | 15.487 |

Table 4: Basic statistics of the PTSLS over 1000 runs when $N = 1000$ and $N = 10000$ in setting (A).

| $N = 1000$ | 1 | $W$ | $X$ |
|---|---|---|---|
| True Coeff. | 1 | 1 | 20 |
| Min. | -33.368 | 2.971 | -113.509 |
| 1st Qu. | -4.258 | 28.662 | 0.518 |
| Median | 2.785 | 45.497 | 28.629 |
| 3rd Qu. | 7.617 | 62.155 | 61.569 |
| Max. | 26.799 | 161.738 | 138.283 |
| Mean | 1.248 | 50.032 | 27.862 |
| SD | 11.374 | 29.523 | 46.388 |

| $N = 10000$ | 1 | $W$ | $X$ |
|---|---|---|---|
| True Coeff. | 1 | 1 | 20 |
| Min. | -12.117 | 30.794 | -20.407 |
| 1st Qu. | -2.336 | 45.907 | 9.404 |
| Median | 1.704 | 50.716 | 19.347 |
| 3rd Qu. | 4.494 | 57.017 | 30.498 |
| Max. | 8.952 | 78.573 | 54.494 |
| Mean | 1.101 | 51.181 | 19.763 |
| SD | 4.638 | 8.814 | 15.171 |

## G.3 ADDITIONAL EXPERIMENTS: NO INTERACTION BETWEEN COVARIATES AND UNOBSERVED CONFOUNDERS

In this section, we give additional experiments with no interaction between covariates and unobserved confounders.

**SCM Settings.** We consider the following two SCMs: $W := H + E_1, X := Z + W + H + E_2$, and

$$\begin{cases} Y := 10X^2 + WX + X + W + 50H + E_3 & \cdots \text{ (C)} \\ Y := \exp(X)\exp(W) + 50H + E_3 & \cdots \text{ (D)} \end{cases} . \tag{108}$$

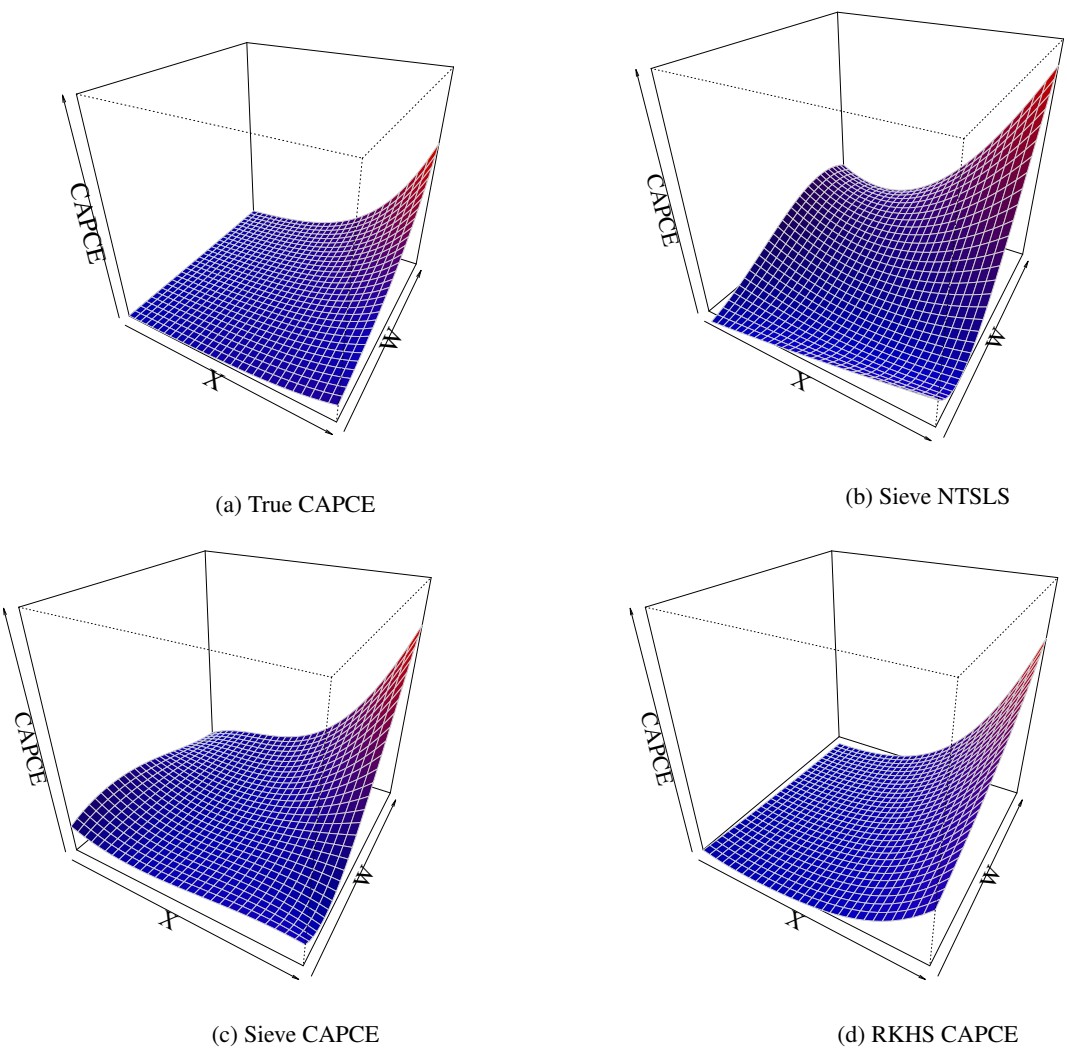

(a) True CAPCE

(b) Sieve NTSLS

(c) Sieve CAPCE

(d) RKHS CAPCE

Figure 5: Nonparametric estimated surfaces in setting (B) (Mean, $N = 10000$). X-axis is the value of treatment variable ($X = x$), Y-axis is the value of covariate ($W = w$), and Z-axis is the value of CAPCE.

The other settings of each estimator are the same as in setting (A) and (B).

**Results.** The basic statistics of estimated coefficients by 100 time simulations of PTSLS and P-CAPCE in setting (C) are shown in Tables 5 and 6. The MSE of each estimator in settings (C) and (D) are shown in Table 7. The results show that the performance of the previous works PTSLS, NTSLS, Kernel IV is comparable with our proposed methods under the settings where the interaction between the covariates $W$ and unobserved confounders $H$ is absent.

Table 5: Basic statistics of the P-CAPCE estimator over 1000 runs when $N = 1000$ and $N = 10000$ in setting (C).

| $N = 1000$ | 1 | $W$ | $X$ |
|---|---|---|---|
| True Coeff. | 1 | 1 | 20 |
| Min. | -4.419 | -38.895 | -21.176 |
| 1st Qu. | 0.896 | 0.351 | 19.614 |
| Median | 0.983 | 1.361 | 19.919 |
| 3rd Qu. | 1.065 | 2.620 | 20.246 |
| Max. | 6.066 | 19.864 | 41.021 |
| Mean | 0.944 | 1.151 | 19.642 |
| SD | 0.811 | 5.535 | 4.884 |

| $N = 10000$ | 1 | $W$ | $X$ |
|---|---|---|---|
| True Coeff. | 1 | 1 | 20 |
| Min. | 0.522 | -21.630 | 18.680 |
| 1st Qu. | 0.976 | 0.158 | 19.901 |
| Median | 1.000 | 0.830 | 20.006 |
| 3rd Qu. | 1.021 | 1.989 | 20.101 |
| Max. | 1.724 | 24.138 | 21.684 |
| Mean | 0.999 | 0.966 | 19.998 |
| SD | 0.106 | 4.851 | 0.305 |

Table 6: Basic statistics of the PTSLS over 1000 runs when $N = 1000$ and $N = 10000$ in setting (C).

| $N = 1000$ | 1 | $W$ | $X$ |
|---|---|---|---|
| True Coeff. | 1 | 1 | 20 |
| Min. | 0.350 | -2.045 | 15.831 |
| 1st Qu. | 0.957 | -0.199 | 19.303 |
| Median | 1.021 | 1.009 | 19.731 |
| 3rd Qu. | 1.101 | 1.921 | 19.953 |
| Max. | 1.456 | 5.165 | 20.260 |
| Mean | 1.029 | 0.997 | 19.474 |
| SD | 0.155 | 1.609 | 0.782 |

| $N = 10000$ | 1 | $W$ | $X$ |
|---|---|---|---|
| True Coeff. | 1 | 1 | 20 |
| Min. | 0.904 | -1.251 | 19.501 |
| 1st Qu. | 0.986 | 0.515 | 19.905 |
| Median | 1.003 | 0.962 | 19.963 |
| 3rd Qu. | 1.021 | 1.407 | 20.019 |
| Max. | 1.074 | 3.148 | 20.120 |
| Mean | 1.003 | 0.939 | 19.939 |
| SD | 0.028 | 0.814 | 0.118 |

Table 7: MSE of estimators in settings (C) and (D).

| MSE | PTSLS | NTSLS | Kernel IV | P-CAPCE | S-CAPCE | RKHS CAPCE |
|---|---|---|---|---|---|---|
| (C) $N = 1000$ | 3.004 | 4.124 | 8.165 | 8.316 | 11.827 | 3.572 |
| (C) $N = 10000$ | 0.254 | 0.367 | 0.614 | 0.774 | 1.324 | 0.567 |
| (D) $N = 1000$ | 139.713 | 4.224 | 4.438 | 52.861 | 4.254 | 2.943 |
| (D) $N = 10000$ | 57.006 | 0.334 | 1.026 | 40.062 | 0.477 | 0.696 |

## G.4 ADDITIONAL EXPERIMENTS: WEAKER INTERACTION BETWEEN COVARIATES AND UNOBSERVED CONFOUNDERS

In this section, we give additional experiments with weak interaction between covariates and unobserved confounders.

**SCM Settings.** We consider the following two SCMs: $W := H + E_1, X := Z + W + H + E_2$, and

$$
\begin{cases}
Y := 10X^2 + WX + X + W + 10(W^5 + W^4 + W^3 + W^2)H + E_3 & \cdots (E) \\
Y := \exp(X)\exp(W) + 5(W^5 + W^4 + W^3 + W^2)H + E_3 & \cdots (F)
\end{cases}. \tag{109}
$$

The other settings of each estimator are the same as in setting (A) and (B).

**Results.** The basic statistics of estimated coefficients by 100 time simulations of PTSLS and P-CAPCE in setting (E) are shown in Tables 8 and 9. The MSE of estimators in settings (E) and (F) are shown in Table 10. The results show that our methods are superior to the previous works PTSLS, NTSLS, Kernel IV while the performance differences are less than that in the settings (A) and (B) where the interaction between covariates and unobserved confounders are stronger.

Table 8: Basic statistics of the P-CAPCE estimator over 1000 runs when $N = 1000$ and $N = 10000$ in setting (E).

| $N = 1000$ | 1 | $W$ | $X$ | | $N = 10000$ | 1 | $W$ | $X$ |
|---|---|---|---|---|---|---|---|---|
| True Coeff. | 1 | 1 | 20 | | True Coeff. | 1 | 1 | 20 |
| Min. | -2.056 | -9.914 | -6.174 | | Min. | 0.057 | -1.633 | 10.648 |
| 1st Qu. | -0.103 | -0.589 | 11.002 | | 1st Qu. | 0.635 | -0.286 | 17.381 |
| Median | 0.848 | 1.996 | 15.528 | | Median | 1.193 | 1.187 | 22.993 |
| 3rd Qu. | 1.898 | 6.510 | 29.150 | | 3rd Qu. | 1.743 | 2.306 | 23.704 |
| Max. | 2.047 | 3.502 | 25.181 | | Max. | 2.047 | 3.502 | 25.181 |
| Mean | 1.185 | 1.765 | 18.248 | | Mean | 1.160 | 0.979 | 20.531 |
| SD | 2.416 | 5.991 | 14.165 | | SD | 0.662 | 1.713 | 4.798 |

Table 9: Basic statistics of the PTSLS over 1000 runs when $N = 1000$ and $N = 10000$ in setting (E).

| $N = 1000$ | 1 | $W$ | $X$ | | $N = 10000$ | 1 | $W$ | $X$ |
|---|---|---|---|---|---|---|---|---|
| True Coeff. | 1 | 1 | 20 | | True Coeff. | 1 | 1 | 20 |
| Min. | -5.345 | -1.064 | -3.732 | | Min. | -1.072 | 7.402 | 11.514 |
| 1st Qu. | -0.074 | 9.187 | 14.958 | | 1st Qu. | 0.394 | 9.613 | 17.370 |
| Median | 1.195 | 10.862 | 20.450 | | Median | 0.957 | 10.748 | 19.355 |
| 3rd Qu. | 2.667 | 13.322 | 24.988 | | 3rd Qu. | 1.446 | 11.977 | 21.967 |
| Max. | 5.766 | 34.620 | 37.757 | | Max. | 2.684 | 15.641 | 26.645 |
| Mean | 1.195 | 10.862 | 20.450 | | Mean | 0.934 | 10.883 | 19.459 |
| SD | 2.161 | 5.437 | 8.597 | | SD | 0.778 | 1.687 | 3.478 |

Table 10: MSE of estimators in settings (E) and (F).

| MSE | PTSLS | NTSLS | Kernel IV | P-CAPCE | S-CAPCE | RKHS CAPCE |
|---|---|---|---|---|---|---|
| (E) $N = 1000$ | 170.152 | 129.132 | 113.733 | 29.408 | 17.181 | 31.267 |
| (E) $N = 10000$ | 64.645 | 55.104 | 65.055 | 3.592 | 4.527 | 3.171 |
| (F) $N = 1000$ | 141.678 | 28.73 | 24.125 | 101.902 | 13.84 | 14.806 |
| (F) $N = 10000$ | 61.569 | 4.323 | 3.197 | 42.422 | 1.788 | 1.731 |

## G.5 ADDITIONAL EXPERIMENTS: ESTIMATION BASED ON THEOREM 3.1'.

In this section, we give additional experiments about estimating CAPCE in the settings (A) and (B) in Eq. (29) based on Theorem 3.1' in Appendix A.2. P-CAPCE', S-CAPCE', and RKHS CAPCE' estimate CAPCE based on Theorem 3.1'. We present detailed settings of numerical experiments in the following.

**Setting of P-CAPCE'.** We learn the conditional expectations of basis functions $\mathbb{E}[Y|Z = z, W = w]$, $\mathbb{E}[X|Z = z, W = w]$, $\mathbb{E}[WX|Z = z, W = w]$ and $\mathbb{E}[X^2|Z = z, W = w]$ by the nonlinear model, $b_0 + b_1 Z + b_2 Z^2$. We used the basis terms $\{1, X, X^2\}$ for P-CAPCE' and $\{1, X\}$ for PTSLS, which match setting (A), and let $z_0 = -1$ and $w = 1$. Regularize value is determined by test error from $\{1, 10^{-1}, 10^{-2}, 10^{-3}\}$.

**Setting of S-CAPCE'.** We learn the conditional expectations by the nonlinear model, $b_0 + b_1 Z + b_2 Z^2 + b_3 Z^3 + b_4 W + b_5 W^2 + b_6 W^3$, We consider the basis terms $h_p(X)$ for $p = 0, 1, 2$, where $h_p$ is Hermite polynomial functions ($h_0(t) = 1$,

$h_1(t) = t$, $h_2(t) = t^2 - 1$ and $h_3(t) = t^3 - 3t$), and let $z_0 = -1$ and $w = 1$. Let $\kappa = 2$ and $l = 1$, and we calculate $\hat{\Lambda}$ by Monte Carlo integration using uniform distribution $x \sim U(-4, 4)$, where $\Omega_X \subseteq [-4, 4]$. Regularize value is determined by test error from $\{1, 10^{-1}, 10^{-2}, 10^{-3}\}$. We estimate CAPCE via differentiating estimated $\mathbb{E}[Y_x | W = w]$.

**Setting of RKHS CAPCE' estimator.** We use polynomial kernel function $k_{Z,W}((z, w)(z, w)^T + C_1)^{C_2}$ and $k_X(x, x') = (xx' + C_3)^{C_4}$. We select the kernel parameters $(C_1, C_2)$ and $(C_3, C_4)$ from $\{1, 2, 3, 4, 5\} \times \{1, 2, 3, 4, 5\}$, respectively. We select the regularize values $\lambda_1$ and $\lambda_2$ from $\{1, 10^{-1}, 10^{-2}, 10^{-3}\}$, respectively, and $(\lambda_3, \xi)$ is from Cartesian product set $\{100, 10, 1\} \times \{100, 10, 1\}$.

**Results.** The MSEs of P-CAPCE, S-CAPCE, RKHS CAPCE, P-CAPCE', S-CAPCE', and RKHS CAPCE' in settings (A) and (B) for $w = 1$ are shown in Table 11. The results show that estimators based on Theorem 3.1 and 3.1' have very similar performance.

Table 11: MSE of each estimator based on Theorem 3.1 and 3.1' in settings (A) and (B) for $w = 1$.

| MSE | P-CAPCE | S-CAPCE | RKHS CAPCE | P-CAPCE' | S-CAPCE' | RKHS CAPCE' |
|---|---|---|---|---|---|---|
| (A) $N = 1000$ | 453.233 | 225.301 | 339.091 | 132.167 | 399.446 | 193.306 |
| (A) $N = 10000$ | 98.885 | 220.358 | 164.798 | 91.5647 | 275.907 | 153.689 |
| (B) $N = 1000$ | 284.598 | 14.398 | 30.562 | 129.721 | 11.780 | 28.266 |
| (B) $N = 10000$ | 52.217 | 5.189 | 3.475 | 63.302 | 5.726 | 3.194 |

## G.6 ADDITIONAL INFORMATION ON THE APPLICATION

We present detailed settings of the application in Section 6. We applied P-CAPCE and PTSLS. We learn the expected values of basis functions by the nonlinear model, $\beta_0 + \beta_1 Z + \beta_2 Z^2$. We use terms $\{1, W, W^2, X, XW, XW^2\}$ for P-CAPCE and $\{1, W, W^2, X, XW, XW^2, X^2, X^2W, X^2W^2\}$ for PTSLS, and let $z_0 = 8$. We estimate CAPCE via differentiating estimated $\mathbb{E}[Y_x | w]$ for PTSLS. Regularize parameter is determined by test error from $\{1, 10^{-1}, 10^{-2}, 10^{-3}, \dots\}$.

**Results.** The basic bootstrapping statistical properties of the P-CAPCE and PTSLS estimators are shown in Tables 12 and 13. The predicted CAPCE values are shown in Tables 14 and 15. The estimated CAPCE surfaces are shown in Fig. 6.

Table 12: Basic statistics of the P-CAPCE estimator over 1000 bootstrapping.

| Terms | 1 | $W$ | $W^2$ | $X$ | $WX$ | $W^2X$ |
|---|---|---|---|---|---|---|
| Min. | -0.00267 | -0.00003 | -0.00126 | -0.00006 | -0.00084 | -0.00237 |
| 1st Qu. | -0.00061 | 0.00002 | 0.00510 | -0.00001 | -0.00029 | -0.00099 |
| Median | -0.00006 | 0.00004 | 0.00904 | -0.00001 | -0.00016 | -0.00053 |
| 3rd Qu. | 0.00058 | 0.00007 | 0.01331 | 0.00000 | -0.00008 | -0.00017 |
| Max. | 0.00226 | 0.00018 | 0.02786 | 0.00004 | 0.00059 | 0.00068 |
| Mean | -0.00003 | 0.00005 | 0.00949 | -0.00001 | -0.00018 | -0.00056 |
| SD | 0.00090 | 0.00004 | 0.00602 | 0.00001 | 0.00018 | 0.00062 |

Table 13: Basic statistics of the PTSLS estimator over 1000 bootstrapping.

| Terms | 1 | $W$ | $W^2$ | $X$ | $WX$ | $W^2X$ |
|---|---|---|---|---|---|---|
| Min. | -0.00082 | -0.03024 | -0.00319 | -0.01294 | -0.47307 | -0.00180 |
| 1st Qu. | 0.00000 | 0.00023 | 0.00729 | 0.00005 | 0.00238 | -0.00126 |
| Median | 0.00007 | 0.00259 | 0.00996 | 0.00134 | 0.06028 | -0.00108 |
| 3rd Qu. | 0.00012 | 0.00499 | 0.01341 | 0.00260 | 0.10789 | -0.00089 |
| Max. | 0.00051 | 0.01539 | 0.04302 | 0.00773 | 0.30066 | 0.00046 |
| Mean | 0.00004 | 0.00191 | 0.01083 | 0.00102 | 0.04417 | -0.00103 |
| SD | 0.00017 | 0.00617 | 0.00660 | 0.00299 | 0.11830 | 0.00039 |

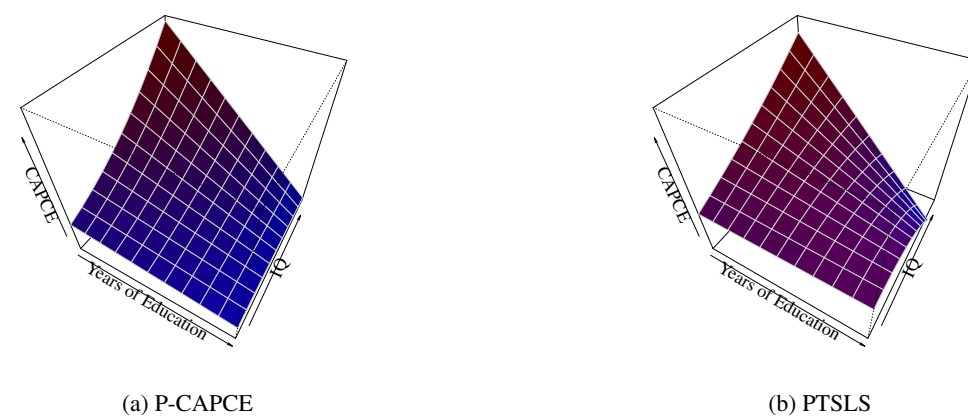

(a) P-CAPCE                                        (b) PTSLS

Figure 6: Bootstrap mean surface of each estimator. X-axis is years of education, Y-axis is IQ, and Z-axis is CAPCE.

Table 14: Predicted CAPCE values by P-CAPCE estimator

| W \ X | 50 | 60 | 70 | 80 | 90 | 100 | 110 | 120 | 130 | 140 | 150 |
|---|---|---|---|---|---|---|---|---|---|---|---|
| 8 | 12.375 | 17.838 | 24.296 | 31.750 | 40.199 | 49.645 | 60.086 | 71.523 | 83.955 | 97.384 | 111.808 |
| 9 | 10.955 | 15.794 | 21.517 | 28.122 | 35.610 | 43.981 | 53.235 | 63.372 | 74.391 | 86.294 | 99.079 |
| 10 | 9.534 | 13.751 | 18.738 | 24.495 | 31.021 | 38.318 | 46.384 | 55.221 | 64.827 | 75.204 | 86.350 |
| 11 | 8.114 | 11.708 | 15.959 | 20.867 | 26.432 | 32.654 | 39.534 | 47.070 | 55.263 | 64.114 | 73.622 |
| 12 | 6.693 | 9.665 | 13.180 | 17.239 | 21.843 | 26.991 | 32.683 | 38.919 | 45.700 | 53.024 | 60.893 |
| 13 | 5.273 | 7.621 | 10.401 | 13.612 | 17.254 | 21.328 | 25.832 | 30.768 | 36.136 | 41.934 | 48.164 |
| 14 | 3.852 | 5.578 | 7.622 | 9.984 | 12.665 | 15.664 | 18.982 | 22.618 | 26.572 | 30.844 | 35.435 |
| 15 | 2.432 | 3.535 | 4.843 | 6.357 | 8.076 | 10.001 | 12.131 | 14.467 | 17.008 | 19.755 | 22.707 |
| 16 | 1.011 | 1.492 | 2.064 | 2.729 | 3.487 | 4.338 | 5.280 | 6.316 | 7.444 | 8.665 | 9.978 |
| 17 | -0.409 | -0.552 | -0.715 | -0.898 | -1.102 | -1.326 | -1.570 | -1.835 | -2.120 | -2.425 | -2.751 |
| 18 | -1.829 | -2.595 | -3.494 | -4.526 | -5.691 | -6.989 | -8.421 | -9.986 | -11.684 | -13.515 | -15.480 |

Table 15: Predicted CAPCE values by PTSLS estimator

| W \ X | 50 | 60 | 70 | 80 | 90 | 100 | 110 | 120 | 130 | 140 | 150 |
|---|---|---|---|---|---|---|---|---|---|---|---|
| 8 | 24.192 | 30.570 | 37.461 | 44.866 | 52.785 | 61.218 | 70.164 | 79.625 | 89.598 | 100.086 | 111.087 |
| 9 | 23.820 | 29.504 | 35.495 | 41.793 | 48.399 | 55.312 | 62.532 | 70.059 | 77.894 | 86.035 | 94.484 |
| 10 | 23.449 | 28.438 | 33.529 | 38.721 | 44.013 | 49.406 | 54.899 | 60.494 | 66.189 | 71.985 | 77.881 |
| 11 | 23.077 | 27.373 | 31.563 | 35.648 | 39.626 | 43.499 | 47.267 | 50.928 | 54.484 | 57.934 | 61.279 |
| 12 | 22.705 | 26.307 | 29.597 | 32.575 | 35.240 | 37.593 | 39.634 | 41.363 | 42.779 | 43.884 | 44.676 |
| 13 | 22.334 | 25.242 | 27.631 | 29.502 | 30.854 | 31.687 | 32.001 | 31.797 | 31.075 | 29.833 | 28.073 |
| 14 | 21.962 | 24.176 | 25.665 | 26.429 | 26.467 | 25.781 | 24.369 | 22.232 | 19.370 | 15.782 | 11.470 |
| 15 | 21.590 | 23.110 | 23.699 | 23.356 | 22.081 | 19.874 | 16.736 | 12.666 | 7.665 | 1.732 | -5.133 |
| 16 | 21.219 | 22.045 | 21.733 | 20.283 | 17.694 | 13.968 | 9.104 | 3.101 | -4.040 | -12.319 | -21.736 |
| 17 | 20.847 | 20.979 | 19.767 | 17.210 | 13.308 | 8.062 | 1.471 | -6.465 | -15.745 | -26.369 | -38.339 |
| 18 | 20.476 | 19.914 | 17.801 | 14.137 | 8.922 | 2.156 | -6.162 | -16.030 | -27.449 | -40.420 | -54.941 |