# OpenReview forum: "Identification and Estimation of Conditional Average Partial Causal Effects via Instrumental Variable"
_auai.org/UAI/2024/Conference — UAI 2024 oral_

### Official Review · Reviewer_QioC · 2024-03-17

**Q2-1 Originality-Novelty:** 3
**Q2-2 Correctness-Technical Quality:** 3
**Q2-5 Clarity Of Writing:** 3

**Q10 Ethical Concerns:**

No.

**Q1 Summary And Contributions:**

The paper extends the new results on the identification and estimation of the average partial causal effect (APCE) to accommodate effect heterogeneity. The paper works under instrumental variable (IV) setting with unmeasured confounding. A weaker identifiability condition was employed and three classes of estimators are proposed along with their theoretical guarantees.

**Q2-3 Extent To Which Claims Are Supported By Evidence:**

3: Good: the main claims are supported by convincing evidence (in the form of adequate experimental evaluation, proofs, (pseudo-)code, references, assumptions).

**Q2-4 Reproducibility:**

4: Excellent: key resources (e.g. proofs, code, data) are available and key details (e.g. proof sketches, experimental setup) are comprehensively described for competent researchers to confidently and easily reproduce the main results.

**Q3 Main Strengths:**

The work offers a clear presentation of the identifiability of conditional APCE with IV. The convergence rates of the estimators proposed are useful. The possibility to investigate the heterogeneity of APCE is valuable in a diverse population.

**Q4 Main Weakness:**

The identification equation (3) is derived under the structural equation model where the IV $Z$ is independent of the conditioning variables $W$. The case that handles the dependence of $Z$ on $W$ gives identification equation (42), which appears to be quite challenging to solve when $W$ contains multiple continuous variables. This was not investigated in the experiments.

**Q5 Detailed Comments To The Authors:**

1. In the second column on page 3, close to the bottom, the constant $\kappa$ is undefined, and Equation (5) appears to have missing parts.
2. In the second column on page 4, about halfway, the symbols for test datasets are inconsistent, such as $\mathcal{D}^{'(1)}$ and $\mathcal{D}^{(1)'}$.
3. How does the choice of the reference level $z_0$ impact the estimator, if at all?
4. How was the mean squared error defined for the simulation studies?
5. The last few sentences of the real data analysis claim that there is a significant difference of APCE between individuals of different IQ levels. But the bootstrap results in Appendix G.5 suggest otherwise, except maybe the coefficient of $W^2X$ from PTSLS.

**Q9 Complying With Reviewing Instructions:**

Yes

---

> ### Author Rebuttal · Authors · 2024-04-04
>
> Thank you for your positive review!
> In the following, your comments are first stated and then followed by our responses.
>
> >Comment:
> The identification equation (3) is derived under the structural equation model where the IV $Z$ is independent of the conditioning variables $W$.
> The case that handles the dependence of $Z$ on $W$ gives identification equation (42), which appears to be quite challenging to solve when $W$ contains multiple continuous variables.
> This was not investigated in the experiments.
>
> Our response:
> We will perform the experiments about estimating CAPCE based on Eq. (42) and add the results in the appendix of the revised version.
>
> >Comment: In the second column on page 3, close to the bottom, the constant $\kappa$ is undefined, and Equation (5) appears to have missing parts.
>
> Our response:
> The constant $\kappa$ can take value that satisfies $\kappa>(1+d)/2$ where $d$ is the dimension of the vector $W$ (specified after Eq. (1)).
> In the experiments, we let $\kappa$ be $2$ and $l$ be $1$ in Eq. (5).
> This compactness restriction is also used in [Newey and Powell, 2003]. We don't find Eq. (5) has missing parts.
>
> >Comment: In the second column on page 4, about halfway, the symbols for test datasets are inconsistent, such as ${\cal D'}^{(1)}$ and $D^{(1)'}$.
>
> Our response:
> We will fix ${\cal D'}^{(1)}$ and ${\cal D'}^{(2)}$ to ${\cal D}^{(1)'}$ and ${\cal D}^{(2)'}$.
>
> >Comment: How does the choice of the reference level $z_0$ impact the estimator, if at all?
>
> Our response:
> The choice of the reference point $z_0$ does not affect the consistency results or rate of convergence, but it may affect the variance of the estimator. In our experiments, we take the minimum value of $Z$ as  a standard reference point $z_0$. The choice of the  reference point $z_0$ did not affect the standard deviation of the estimators much in our experiments.
>
> >Comment: How was the mean squared error defined for the simulation studies?
>
> Our response:
> MSE is computed as  $\frac{1}{N_1'}\sum_{i=1}^{N_1'}(\hat{g}(x_i^{(1)'},w_i^{(1)'})-g(x_i^{(1)'},w_i^{(1)'}))^2$ with test dataset ${\cal D}^{(1)'}$.
>
> >Comment: The last few sentences of the real data analysis claim that there is a significant difference of APCE between individuals of different IQ levels. But the bootstrap results in Appendix G.5 suggest otherwise, except maybe the coefficient of $W^2X$ from PTSLS.
>
> Our response:
> Although the coefficients in Tables 11 and 12 appear to be small, when plugging in the IQ values, e.g. $W=80, W=120$, the differences in APCE values are significant.
> For instance, Table 13 shows that for students of 8 years of education, APCE is $31.750$ for IQ 80  and  is $71.523$ for IQ 120, a significant difference.

---

### Official Review · Reviewer_ibvE · 2024-03-19

**Q2-1 Originality-Novelty:** 3
**Q2-2 Correctness-Technical Quality:** 3
**Q2-5 Clarity Of Writing:** 4

**Q1 Summary And Contributions:**

The paper studies the identification of conditional average partial causal effects (CAPCE) in the form of $E[\partial_xY_x|\mathbf{w}]$ in the instrumental variable (IV) model. The estimation of CAPCE is centered around an integral equation that connects CAPCE with other quantities available from observational distributions. In particular, the paper proposes a separability assumption for estimating CAPCE which is weaker than the assumption for identifying $E[Y_x|\mathbf{w}]$ in [Newey and Powell 2003]  (note that $E[\partial_xY_x|\mathbf{w}]$ is always estimable from $E[Y_x|\mathbf{w}]$). It also extends the previous results in [Wong, 2022] and [Kawakami et al. 2023] which concern $E[\partial_xY_x]$ without covariates $\mathbf{W}$. The paper provides three estimation methods: Sieve CAPCE estimator (non-parametric), parametric CAPCE estimator, and reproducing kernel Hilbert space estimator. Theoretical properties of these estimators are studied, and experiments are conducted to demonstrate the effectiveness of these estimators under the weaker separability assumption.

**Q2-3 Extent To Which Claims Are Supported By Evidence:**

4: Excellent: all claims are supported by very convincing evidence (in the form of comprehensive experimental evaluation, rigorous mathematical proofs, detailed (pseudo-)code, precise references, well-motivated and realistic assumptions) and the authors deliver what they promise.

**Q2-4 Reproducibility:**

3: Good: key resources (e.g. proofs, code, data) are available and key details (e.g. proofs, experimental setup) are sufficiently well-described for competent researchers to confidently reproduce the main results.

**Q3 Main Strengths:**

- The problem of identifying and estimating CAPCE under the IV model is novel and significant.
- The paper is well structured and clearly written.
- The statistical behaviors of the estimators (consistency and rate of convergence) were studied. Moreover, empirical studies on both simulation (Section 5) and real-world dataset (Section 6) were provided to illustrate the superiority of the proposed estimators.

**Q4 Main Weakness:**

- More background on reproducing kernel Hilbert space (RKHS) will be very helpful to those not familiar with these math concepts.
- The theorems (Thm 4.1, 4.2, 4.3, 4.4, 4.5) on the statistical properties of estimators contain many assumptions. While most of these assumptions were already known from [Newey and Powell, 2003], [Ai and Chen 2003], adding more intuitions/examples can help illustrate the extent of their restrictiveness.

**Q5 Detailed Comments To The Authors:**

1. I think it'll be more convincing if the paper can include a concrete example where the taking the derivative of $E[Y_x|\mathbf{w}]$ computed from (Newey and Powell 2003) yields an incorrect answer when the weaker separability assumption (Assumption 3.2) holds but the previous separability assumption (Newey and Powell 2003) does not.

2. Are there any restrictions on the functions $\theta_k$ in the parametric CAPCE estimator? For example, do they need to be finite?

3. For RKHS CAPCE estimator, are $\lambda_1 ||G_1||^2_{L2(HZ, HX, W)}$, $\lambda_2||G_2||^2$, ... in Eq. (20), (21), (22) regularization terms? Any reason why these specific terms are considered?

4. Before spelling out the details of two stages of Sieve CAPCE estimator, it would be helpful to mention the high-level idea of each stage.

5. Right above equation (5), "$\kappa > (1+d) / 2$, and define regulaized Sobolev norm..." I don't think $d$ is defined here. Is $d = |\mathbf{w}|$?

**Q9 Complying With Reviewing Instructions:**

Yes

---

> ### Author Rebuttal · Authors · 2024-04-03
>
> Thank you for your constructive comments and suggestions. They are  helpful for us to improve our paper. We will carefully incorporate them in the revised paper. In the following, your comments are first stated and then followed by our responses.
>
> >Comment:
> More background on reproducing kernel Hilbert space (RKHS) will be very helpful to those not familiar with these math concepts.\\
> The theorems (Thm 4.1, 4.2, 4.3, 4.4, 4.5) on the statistical properties of estimators contain many assumptions. While most of these assumptions were already known from [Newey and Powell, 2003], [Ai and Chen 2003], adding more intuitions/examples can help illustrate the extent of their restrictiveness.
>
> Our response:
> Thanks for the feedback. We will add more explanations.
>
> >Comment: I think it'll be more convincing if the paper can include a concrete example where the taking the derivative of $E[Y_x|w]$ computed from (Newey and Powell 2003) yields an incorrect answer when the weaker separability assumption (Assumption 3.2) holds but the previous separability assumption (Newey and Powell 2003) does not.
>
> Our response:
> We believe the examples and experiments presented in Section 5 served this purpose. The two SCMs (A) and (B) in Eq. (27) satisfy the weaker separability assumption (Assumption 3.2) but not the separability assumption (Eq. (2)) required by the previous work. The results in Table 1 shows that the estimated
> coefficients of P-CAPCE are converging to the true values
> when the sample size $N = 10000$, while the coefficient for $W$ estimated by previous method PTSLS is still biased.
> The results in Table 2 shows that the MSE of the previous methods (PTSLS, NTSLS, Kernel IV) are larger than our corresponding methods.
>
> We further performed experiments in settings where the strong separability holds (discussed in the last paragraph in Section 5), and the results (presented in Appendix G) show that the performances
> of the existing methods
> are comparable with our proposed methods under this situation.
>
> >Comment:
> Are there any restrictions on the functions in the parametric CAPCE estimator? For example, do they need to be finite?
>
> Our response:
> As a standard regression, we'd  choose a set of linearly independent functions.
> The number of functions are assumed to be finite.
> To use infinite numbers of functions, we should resort to the Sieve estimator.
>
>
> >Comment:
> For RKHS CAPCE estimator, are
> $\lambda_1\|G_1\|^2$, $\lambda_2\|G_2\|^2$, ... in Eq. (20), (21), (22) regularization terms? Any reason why these specific terms are considered?
>
> Our response:
> Yes, $\lambda_1\|G_1\|^2$, $\lambda_2\|G_2\|^2$, ... in Eq. (20), (21), (22) are regularization terms.
> These regularization terms restrict the $L_2$ norm of models, and these are standard regularization for kernel ridge regression.
>
>
> >Comment:
> Before spelling out the details of two stages of Sieve CAPCE estimator, it would be helpful to mention the high-level idea of each stage.
>
> Our response:
> Thanks for the feedback.
> We will add explanations along the line of the following sentences:
> ``In stage 1, we learn models $\hat{E}[Y|Z=z]$ and
> $\hat{E}[\varphi_j(X,W)|Z=z]$  from the datasets by regression. Then in Stage 2, we estimate parameters ${\boldsymbol \beta}$ by solving Eq. (7).''
>
> >Comment:
> Right above equation (5), "$\kappa>(1+d)/2$, and define regularized Sobolev norm..." I don't think $d$ is defined here. Is $d=|w|$?
>
> Our response:
> Yes, $d=|w|$. The paper mentions $W$ is $d$-dimensional after Eq. (1); we will add $d=|w|$ before Eq. (5).

---

### Official Review · Reviewer_Yf8d · 2024-03-20

**Q2-1 Originality-Novelty:** 2
**Q2-2 Correctness-Technical Quality:** 2
**Q2-5 Clarity Of Writing:** 2

**Q1 Summary And Contributions:**

The paper studies the setting with a continuous treatment, instrumental variable, and covariate. The goal is to identify and estimate the expected derivative of the counterfactual outcome with respect to treatment, conditional on covariates.

**Q2-3 Extent To Which Claims Are Supported By Evidence:**

2: Fair: the main claims are somewhat supported by evidence (but the experimental evaluation may be weak, or does not match entirely with the claims, important baselines may be missing, proofs contain important ideas but lack rigor, algorithmic details are only discussed superficially, references are imprecise, assumptions are not sufficiently motivated or explicated, etc.).

**Q2-4 Reproducibility:**

3: Good: key resources (e.g. proofs, code, data) are available and key details (e.g. proofs, experimental setup) are sufficiently well-described for competent researchers to confidently reproduce the main results.

**Q3 Main Strengths:**

The main strengths are the identification and estimation results, which extend previous results without the covariate.

**Q4 Main Weakness:**

1. The abstract and beginning of section 3 emphasize that this paper “introduces a new concept, conditional average partial causal effect (CAPCE)”. I find this claim to be an unnecessary overstatement.

1a. For example, in statistics, it would be called the partial derivative of the variable importance curve. In econometrics, it would be called the partial derivative of the nonparametric instrumental variable regression in the partially endogenous setting. There are more names.

1b. A more accurate claim would be that this paper studies CAPCE with unobserved confounding yet without separability, by introducing a new model.

2. The abstract and remark of section 3 emphasize that this paper’s identifying assumptions are “weaker than the assumption needed by existing work” for the nonparametric instrumental variable regression function with covariates. I agree that this paper avoids the separability assumption, but I am not sure the characterization of the previous separability assumptions, given as (2) in Section 2, is correct.

2a. Consider Figure 1 and display (1). This DAG differs from what is typically meant by instrumental variables with baseline covariates. What is typically meant is that W has no arrows into it, and W has arrows out of it pointing to Z, X, and Y. Such a DAG is the partially endogenous setting in Newey and Powell (2003) and related work, which this paper holds up for comparison.

2b. Clearly Figure 1 is something different. It may be a valid model to study, but the comparison does not seem right, and was a source of confusion for me. It seems this paper avoids separability but also changes the problem by changing the role of the covariates. Is the absent arrow from W to Z crucial to the identification argument? It is a stronger requirement on the instrument. On the other hand, allowing an arrow from H to W is a weaker requirement on the covariate. These differences should be discussed.

2c. Within the context of the model of Figure 1, I do not see how the separability assumption stated as (2) implies the identifying expression stated immediately before the display.

3. The title does not mention instrumental variables, which is where this paper’s contributions for CAPCE lie.

**Q5 Detailed Comments To The Authors:**

I will raise the score if these items are improved:
1) the presentation of the covariate model;
2) the discussion of separability;
3) the framing of CAPCE being new versus this model for CAPCE being new.

**Q9 Complying With Reviewing Instructions:**

Yes

---

> ### Author Rebuttal · Authors · 2024-04-03
>
> Thank you for your constructive comments and suggestions. They are  helpful for us to improve our paper. We will carefully incorporate them in the revised paper. In the following, your comments are first stated and then followed by our responses.
>
> >Comment:
> The abstract and beginning of section 3 emphasize that this paper “introduces a new concept, conditional average partial causal effect (CAPCE).”
> I find this claim to be an unnecessary overstatement.\
> 1a. In statistics, it is called the partial derivative of the variable importance curve. In econometrics, it is called the partial derivative of the nonparametric instrumental variable regression in the partially endogenous setting. There are more names.\
> 1b. A more accurate claim is that this paper studies CAPCE with unobserved confounding yet without separability, by introducing a new model.
>
> Our response:
> Thanks for the feedback. We will replace the sentence in the abstract "In this paper, we introduce conditional average partial causal effects (CAPCE)" with "In this paper, we study conditional average partial causal effects (CAPCE)." We will replace the first sentence in Section 3 "First, we introduce a new concept conditional average partial causal effect (CAPCE) to capture the heterogeneous causal effects of a continuous treatment" with "First, we formally define  conditional average partial causal effect (CAPCE) to capture the heterogeneous causal effects of a continuous treatment." We agree that the quantity represented by CAPCE has been implicitly studied in the literature. Still we think it's important to formally define and name this quantity.
>
> >Comment:
> The abstract and remark of section 3 emphasize that this paper’s identifying assumptions are “weaker than the assumption needed by existing work” for the nonparametric instrumental variable regression function with covariates.
> I am not sure the characterization of the previous separability assumptions is correct.\
> 2a. Figure 1 differs from what is typically meant by instrumental variables with baseline covariates. What is typically meant is that $W$ has no arrows into it, and $W$ has arrows out of it pointing to $Z$, $X$, and $Y$.
> Such a DAG is the partially endogenous setting in Newey and Powell (2003) and related work, which this paper holds up for comparison.\
> 2b. Figure 1 is something different. It may be a valid model to study, but the comparison does not seem right, and was a source of confusion for me. It seems this paper avoids separability but also changes the problem by changing the role of the covariates. Is the absent arrow from $W$ to $Z$ crucial to the identification argument? It is a stronger requirement on the instrument. On the other hand, allowing an arrow from $H$ to $W$ is a weaker requirement on the covariates. These differences should be discussed.
>
> Our response:
>
> -Figure 1 is also a popular model for studying IV with covariates, e.g., in  [Huntington-Klein, 2020] and the following two papers:\
> (1) Wu, A., et al. (2022). Instrumental variables in causal inference and machine learning: A survey.\
> (2) Hartford, J., et al. (2017). Deep IV: A flexible approach for counterfactual prediction. In International Conference on Machine Learning. PMLR.\
> (It seems Figure 1 is popular in the machine learning literature while economists typically allow the arrow from $W$ to $Z$ though they often don't use DAGs.)
>
> -We have discussed the IV model with an arrow from $W$ to $Z$ (Figure 3 in the Appendix) at the end of the Conclusion section with results given in Appendix A.2. CAPCE is still identifiable under Figure 3 (with the same assumption for identifying CAPCE under Figure 1). But estimating CAPCE under Figure 3 is more difficult as under Figure 3, we have to learn CAPCE=$E[\partial_x Y_{x}|w]$ as a function of $x$ for each $w \in \Omega_W$ respectively, while under Figure 1,  we can learn $E[\partial_x Y_x|w]$ directly as a function of $x$ and $w$.
>
> -As you concerns, if there exists the arrow from $W$ to $Z$ (Figure 3 in Appendix) the conditions of Newey and Powell (2003) become $f_Y(X,W,H,u_Y)=f_Y^1(X,W,u_Y)+f_Y^2(H,u_Y)$ and
> $E[f_Y^2(H,u_Y)|Z,W]=0$.
> According to Theorem 3.1' in the Appendix, we can identify CAPCE under Fig 3 with the same assumptions of Theorem 3.1.
> Thus, our identification assumptions are weaker than the assumptions needed by existing work, even in Figure 3.
>
> >Comment: Within the context of the model of Figure 1, I do not see how the separability assumption stated as (2) implies the identifying expression stated immediately before the display.
>
> Our response: This result is from [Newey and Powell, 2003]. In their notation, the separability conditions are $Y=f_Y^1(X,W)+e$ and
> $E[e|Z,W]=0$.
>
> >Comment:
> The title does not mention instrumental variables, which is where this paper’s contributions for CAPCE lie.
>
> Our response: We plan to  change our title to
> "Identification and Estimation of Conditional Average Partial Causal Effects via Instrumental Variable."

---

### Official Review · Reviewer_bbF3 · 2024-03-22

**Q2-1 Originality-Novelty:** 3
**Q2-2 Correctness-Technical Quality:** 3
**Q2-5 Clarity Of Writing:** 3

**Q1 Summary And Contributions:**

The paper derives conditions for identifiability and inference over an estimand CAPCE (Conditional Average Partial Causal Effect).
To the best of my knowledge, it develops the recent research agenda on APEs and IVs, where Wong (2021) is one of the notable examples.
In this sense, they provide interesting contribution.

**Q2-3 Extent To Which Claims Are Supported By Evidence:**

3: Good: the main claims are supported by convincing evidence (in the form of adequate experimental evaluation, proofs, (pseudo-)code, references, assumptions).

**Q2-4 Reproducibility:**

3: Good: key resources (e.g. proofs, code, data) are available and key details (e.g. proofs, experimental setup) are sufficiently well-described for competent researchers to confidently reproduce the main results.

**Q3 Main Strengths:**

I believe the paper is very detailed, well-written clear.

**Q4 Main Weakness:**

No clear main weakness.

**Q5 Detailed Comments To The Authors:**

I believe the paper is well-written and clear. First of, it provides a succinct but adequate discussion of the literature on instrumental variables as well as the recent results using a variety of parametric restrictions (separability, etc).

I would like to see other points being discussed as well. For instance, the paper is discussed from the point of instrumental variables, but CAPCE have importance for other scenarios as well. For instance, the estimand is important when one does not have noncompliance, although the identification challenges would be somewhat easier.  Anyway, this could be expanded to show the importance of the estimand.

**Q9 Complying With Reviewing Instructions:**

Yes

---

> ### Author Rebuttal · Authors · 2024-04-03
>
> Many thanks for your positive review!
>
> >Comment:
> I would like to see other points being discussed as well.
> For instance, the paper is discussed from the point of instrumental variables, but CAPCE have importance for other scenarios as well.
> For instance, the estimand is important when one does not have noncompliance, although the identification challenges would be somewhat easier.
> Anyway, this could be expanded to show the importance of the estimated.
>
> Our response:
> Yes, the quantity represented by CAPCE has applications in scenarios beyond discussed in the paper, e.g. in situations without noncompliance. Thanks for pointing us to potential new research directions.

---

### Official Review · Reviewer_rynB · 2024-03-27

**Q2-1 Originality-Novelty:** 3
**Q2-2 Correctness-Technical Quality:** 3
**Q2-5 Clarity Of Writing:** 3

**Q10 Ethical Concerns:**

There is no ethical concern.

**Q1 Summary And Contributions:**

This paper introduces conditional average partial causal effect (CAPCE) to represent the heterogeneous causal effects of a continuous treatment. It is showed that CAPCE is identifiable under a weaker assumption than required by $E(Y_x|W)$. This paper also develops three families of CAPCE estimators: sieve, parametric, and RKHS, and analyzes their statistical properties. Experiment results show that the proposed CAPCE estimators are superior to several existing widely used IV methods in settings where the standard separability assumption is violated.

**Q2-3 Extent To Which Claims Are Supported By Evidence:**

3: Good: the main claims are supported by convincing evidence (in the form of adequate experimental evaluation, proofs, (pseudo-)code, references, assumptions).

**Q2-4 Reproducibility:**

3: Good: key resources (e.g. proofs, code, data) are available and key details (e.g. proofs, experimental setup) are sufficiently well-described for competent researchers to confidently reproduce the main results.

**Q3 Main Strengths:**

This paper shows that CAPCE is identifiable under a weaker assumption than the standard separability assumption.  This paper proposes three estimators for CAPCE clearly. The statistical properties of these proposed estimators are also studies, which is theoretically solid. Extensive simulation studies are conducted to compare the proposed new estimators with several existing estimators. The proposed estimators have some advantages over existing estimators in some cases.

**Q4 Main Weakness:**

The proposed RKHS CAPCE method is very time consuming.

The estimated value of coefficient for W is far from the true coefficient for the P-CAPCE method when the sample size is 1000.

In the real-data example, the sample size is 857, which is not too small, but the authors still say that ``We applied P-CAPCE and PTSLS. Other estimators are not used due to the small sample size." The proposed S-CAPCE and RKHS CAPCE methods have high requirement on the sample size, which may not be applicable in many practical problems.

There are some typo errors in this paper.

There are some points that need to be explained and clarified.

**Q5 Detailed Comments To The Authors:**

What are the meanings of other curves in Figure 2(a)?

In Theorem 4.5, it states that the RKHS CAPCE estimator converges pointwise to CAPCE when \lambda_3=0, can we set \lambda_3 to be zero in the optimization problem (22)?

There maybe some typo errors. For example, in the definition of \mu(z) in Theorem 3.1, should we swap  the positions of z_0 and z? In Equation (22), do we need to swap the positions of G_1 and G_2?

The true CAPCE depends on the values of x and w, how do you get the overall MSE for all true CAPCE values?

**Q9 Complying With Reviewing Instructions:**

Yes

---

> ### Author Rebuttal · Authors · 2024-04-03
>
> Thank you for your constructive comments and suggestions. They are helpful for us to improve our paper. We will carefully incorporate them in the revised paper. In the following, your comments are first stated and then followed by our responses.
>
> >Comment:
> What are the meanings of other curves in Figure 2(a)?
>
> Our response:
> The dot-dashed curves around the P-CAPCE and PTSLS curves are 95\% confidence interval curves.
>
> >Comment:
> In Theorem 4.5, it states that the RKHS CAPCE estimator converges pointwise to CAPCE when $\lambda_3=0$, can we set $\lambda_3$ to be zero in the optimization problem (22)?
>
> Our response: No, when $\lambda_3=0$, the inverse of the matrix $\hat{O}\hat{O}^T+N_2\xi {K}_{(X,W)^{(1)}(X,W)^{(1)}}$ in Eq. (24) is numerically unstable.
> Regularization leads to bias, but we must consider the bias-variance trade-off in practice.
> This is a common problem for kernel methods.
>
> >Comment:
> There maybe some typo errors.
> For example, in the definition of $\mu(z)$ in Theorem 3.1, should we swap the positions of $z_0$ and $z$? In Equation (22), do we need to swap the positions of $G_1$ and $G_2$?
>
> Our response: These are not typos; the  equations in the paper are correct. As an illustration, the following derivation shows how the positions of $z$ and $z_0$ are swapped from Eq. (3) when substituting $g(x)=\sum_{j=1}^J\beta_j\phi_j(x)$ into Eq. (3) without covariates $W$. We have
> $$E[Y|Z=z_0]-E[Y|Z=z]=\int_{\Omega_X} \{p(X\leq x|Z=z)-p(X\leq x|Z=z_0)\}\sum_{j=1}^J\beta_j\phi_j(x) dx$$
> $$\Leftrightarrow E[Y|Z=z_0]-E[Y|Z=z]=\sum_{j=1}^J\beta_j\int_{\Omega_X} \{p(X> x|Z=z_0)-p(X> x|Z=z)\}\phi_j(x) dx$$
> $$\Leftrightarrow E[Y|Z=z_0]-E[Y|Z=z]
>     =\sum_{j=1}^J\beta_j\{E[\varphi_j(X)|Z=z_0]-E[\varphi_j(X)|Z=z]\}$$
> $$\Leftrightarrow E[Y|Z=z]-E[Y|Z=z_0]=\sum_{j=1}^J\beta_j\{E[\varphi_j(X)|Z=z]-E[\varphi_j(X)|Z=z_0]\}.$$
> $G_1$ is the model to predict $Y$ based on $Z$, $G_2$ is the model to predict $\pi(X,W)$ based on $Z$.
> Thus, Eq. (22) is consistent with (3).
>
> >Comment:
> The true CAPCE depends on the values of $x$ and $w$, how do you get the overall MSE for all true CAPCE values?
>
> Our response: MSE is computed as  $\frac{1}{N_1'}\sum_{i=1}^{N_1'}(\hat{g}(x_i^{(1)'},w_i^{(1)'})-g(x_i^{(1)'},w_i^{(1)'}))^2$ with test dataset ${\cal D}^{(1)'}$.

---

### Meta-Review · Area_Chair_vSVF · 2024-04-18

Four out of five reviewers contend that this paper contributes to the literature on assessing the effects of continuous treatments and should be accepted. The reviewers identify some previous literature on closely related topics. Addressing these changes seems reasonable  within the UAI process.


One follow-up question:

Reviewer rynB asked:
What are the meanings of other curves in Figure 2(a)?
The authors responded: The dot-dashed curves around the P-CAPCE and PTSLS curves are 95% confidence interval curves.

Could you clarify whether these are confidence bands (that cover the whole curve with confidence 95%) or are they just pointwise confidence intervals.